# MODEL-AGNOSTIC META-LEARNERS FOR ESTIMATING HETEROGENEOUS TREATMENT EFFECTS OVER TIME

**Dennis Frauen,**[*] **Konstantin Hess & Stefan Feuerriegel**
LMU Munich
Munich Center of Machine Learning (MCML)
`{frauen,hess,feuerriegel}@lmu.de`

## ABSTRACT

Estimating heterogeneous treatment effects (HTEs) over time is crucial in many disciplines such as personalized medicine. Existing works for this task have mostly focused on model-based learners that adapt specific machine-learning models and adjustment mechanisms. In contrast, model-agnostic learners – so-called *meta-learners* – are largely unexplored. In our paper, we propose several meta-learners that are model-agnostic and thus can be used in combination with arbitrary machine learning models (e.g., transformers) to estimate HTEs over time. We then provide a comprehensive theoretical analysis that characterizes the different learners and that allows us to offer insights into when specific learners are preferable. Furthermore, we propose a novel IVW-DR-learner that (i) uses a doubly robust (DR) and orthogonal loss; and (ii) leverages inverse-variance weights (IVWs) that we derive to stabilize the DR-loss over time. Our IVWs downweight extreme trajectories due to *products* of inverse-propensities in the DR-loss, resulting in a lower estimation variance. Our IVW-DR-learner achieves superior performance in our experiments, particularly in regimes with low overlap and long time horizons.

## 1 INTRODUCTION

Estimating heterogeneous treatment effects (HTEs) from observational data over time is crucial when randomized experiments are costly or infeasible (Lim et al., 2018; Bica et al., 2020a; Melnychuk et al., 2022). A typical example is personalized medicine (Feuerriegel et al., 2024). Here, medications are commonly prescribed over the course of several days, and it is nowadays standard that patient responses are captured within electronic health records (Allam et al., 2021). Hence, clinicians can use electronic health records to estimate HTEs over time and thereby analyze the safety and effectiveness of treatments across different patient profiles (Hamburg & Collins, 2010). For example, in cancer care, clinicians are commonly interested in which sequence of chemotherapy and immunotherapy has the largest expected benefit for patients with a specific age, gender, or medical history.

Estimating HTEs *over time* (also referred to as the *time-varying setting*) poses unique challenges due to the complex underlying data-generating process and time-varying confounding structure. In particular, estimation is hard because of the following three challenges: (i) **Time-varying confounding**: the existence of post-treatment confounders biases naïve estimators and requires adjustment mechanisms tailored to the time-varying setting (Robins et al., 2000; van der Laan & Gruber, 2012). (ii) **Growing history**: this means that one must adjust for entire patient histories growing over time (Bica et al., 2020a). (iii) **Low overlap:** sequences of treatments are less likely observed in observational data from time-varying settings (as compared to single treatments in static settings), which thus increases the variance of estimators (Hess et al., 2024b). This problem increases with the length of the time-horizon.

Existing works for estimating HTEs over time are almost entirely *model-based* (see Table 1). That is, these works propose specific model architectures and/or losses designed to address the challenges (i)–(iii) from above, or at least some of them (Lim et al., 2018; Bica et al., 2020a; Melnychuk et al., 2022). Furthermore, these model-based learners build upon a certain set of adjustment mechanisms (e.g., history adjustment, inverse propensity weighting). In contrast, other important adjustment

---

[*]Corresponding author

mechanisms such as doubly robust adjustment (which typically offers favorable theoretical properties) remain overlooked. This is surprising, since model-agnostic methods – so-called *meta-learners* – are commonly used in the static setting (i.e., without time dimension) and implemented popular in software packages (Battocchi et al., 2019), whereas a comprehensive toolbox of meta-learners for the time-varying setting is still missing.

In this paper, we take a model-agnostic view on estimating HTEs over time. To this end, we propose several *meta-learners* for this task that can be used in combination with arbitrary machine-learning models and come with theoretical guarantees such as Neyman-orthogonality and double robustness. Our meta-learners address the challenges (i)–(iii) from above as follows: (i) they leverage adjustment strategies tailored to the time-series setting to ensure unbiased estimation, including regression adjustment (G-computation), propensity adjustment, and doubly robust adjustment; (ii) they can be instantiated using state-of-the-art models such as transformers to effectively learn representations from high-dimensional, time-varying data; and (iii) we show that certain meta-learners (e.g., the doubly robust learner) can suffer from high variance under violations of overlap due to division by propensity scores (or products thereof). As a remedy, we derive time-varying inverse-variance weights to stabilize the doubly robust loss, which yields the inverse-variance-weighted doubly robust learner (called IVW-DR-learner).

Our **contributions**[1] are: (1) We propose several meta-learners for estimating HTEs over time by leveraging various adjustment mechanisms. They are model-agnostic; i.e., they can be instantiated using any machine learning model of choice such as transformers. (2) We analyze the meta-learners theoretically and provide insights for when specific learners are preferable over others. We also structure the literature by showing that some existing model-based learners are instantiations of certain meta-learners, while model-based learners corresponding to other meta-learners have not been proposed yet (see Tables 1 and 5). (3) We derive inverse variance weights to stabilize the loss of the DR-learner, yielding a novel IVW-DR-learner. We also perform various experiments to confirm the effectiveness of our IVW-DR-learner as well as the validity of our theoretical results.

## 2 RELATED WORK

Related works on estimating HTEs can be grouped along two dimensions (see Table 1): (a) whether a method is model-based or model-agnostic, and (b) whether a method is for the static or time-varying setting.

**Learners in the static setting:** Both (i) model-based and (ii) model-agnostic meta-learners have been proposed in the static setting, but without considering the time dimension. (i) Prominent model-based

Table 1: Key learners for heterogeneous treatment effects.

|  | Static setting | Time-varying setting |
|---|---|---|
| Model-based | CFR-Net (Johansson et al., 2016), TARNet (Shalit et al., 2017), Causal forest (Wager & Athey, 2018), etc. | RMSNs (Lim et al., 2018), CRN (Bica et al., 2020a), CT (Melnychuk et al., 2022) G-Net (Li et al., 2021), GT (Hess et al., 2024a) |
| Model-agnostic | Plug-in-, RA-, IPW (Curth & van der Schaar, 2021)-, DR (Kennedy, 2023)-, IVW-DR- (Fisher, 2024)-, R-learner (Nie & Wager, 2021) | R-learner (Lewis & Syrgkanis, 2021), PI-HA-, PI-RA-, RA-, IPW-, DR-, IVW-DR-learner |

learners include the causal forest (Wager & Athey, 2018) and TARNet (Shalit et al., 2017). (ii) Meta-learners in the static setting leverage different adjustment mechanisms. Examples include the RA- or X-learner (Künzel et al., 2019; Curth & van der Schaar, 2021), the IPW-learner (Curth & van der Schaar, 2021), the DR-learner (Kennedy, 2023), the R-learner (Nie & Wager, 2021), the EP-learner (van der Laan et al., 2024), and the recently proposed inverse-variance-weighted DR-leaner (IVW-DR-learner) (Fisher, 2024). Note that these learners focus on the static setting (and not the time-varying setting). As a result, these are *inherently biased for estimating HTEs over time* because runtime confounding renders static adjustment mechanisms insufficient. Meta-learners are often based on semiparametric theory and come with theoretical guarantees (Chernozhukov et al., 2018; Foster & Syrgkanis, 2023). Despite being well-established in the static setting (Acharki et al., 2023; Curth & van der Schaar, 2021; Kennedy, 2023; Nie & Wager, 2021), similar learners for the time-series setting are missing.

**Model-based learners for HTEs in the time-varying setting:** Several model-based learners have been proposed for estimating HTEs over time. These learners leverage (i) specific model architec-

---
[1]Code is available at https://github.com/DennisFrauen/CATEMetaLearnersTime.

tures (e.g., recurrent neural networks or transformers with balancing representations (Bica et al., 2020a; Hess et al., 2024a; Melnychuk et al., 2022)) or (ii) specific adjustment mechanisms (e.g., G-computation (Li et al., 2021) or inverse propensity weighting (Lim et al., 2018)). However, these learners are *not* model agnostic. To the best of our knowledge, none of these learners uses a doubly robust adjustment mechanism. Further, a theoretical analysis comparing the different learners is missing.

**Meta-learners for HTEs in the time-varying setting:** We are only aware of a single meta-learner for the time-varying setting: the R-learner proposed in (Lewis & Syrgkanis, 2021). While it is model-agnostic, it imposes parametric assumptions on the data-generating process (e.g., linear Markov assumptions). In contrast, our meta-learners are completely nonparametric. We provide a detailed comparison of our meta-learners with the R-learner in Table 5 and Appendix B.

**Meta-learners in reinforcement learning (RL):** The time-varying setting for estimating HTEs can be viewed as a model-free off-policy learning problem in RL, where we are interested in estimating advantage functions (A-learning (Murphy, 2003)) in a non-Markovian decision process (NMDP) (Uehara et al., 2022). Recently, meta-learners have been proposed to estimate the average reward of a policy in NMDPs (Kallus & Uehara, 2019; 2020; 2022). However, these works differ from ours in two ways: (i) they do consider a different estimand and are not directly applicable for advantage functions/ HTEs; and (ii) to the best of our knowledge, no previous work in RL has leveraged inverse variance weights to deal with the widespread problem of low overlap.

## 3 PROBLEM SETUP

**Data-generating process:** We build upon the standard setting for estimating HTEs from observational data over time (Bica et al., 2020a; Melnychuk et al., 2022). That is, we consider covariates $X_t \in \mathcal{X}$, discrete treatments $A_t \in \mathcal{A}$, and continuous outcomes $Y_t \in \mathbb{R}$ that are observed over several time periods $t$. In medical settings, $X_t$ may denote (time-varying) patient characteristics such as heart rate and blood pressure, $A_t$ may denote whether a drug is administered, and $Y_t$ may denote a health outcome such as the blood sugar level.

For each time period $t$, the treatment $A_t$ is assigned based on the history $\bar{H}_t = \{\bar{X}_t, \bar{A}_{t-1}, \bar{Y}_{t-1}\}$, where $\bar{X}_t = (X_1, \ldots, X_t)$, $\bar{Y}_{t-1} = (Y_1, \ldots, Y_{t-1})$, and $\bar{A}_{t-1} = (A_1, \ldots, A_{t-1})$. The outcome $Y_t$ depends on both the history $\bar{H}_t$ and the assigned treatment $A_t$. Furthermore, the time-varying covariates $X_t$ (e.g., blood pressure) and treatments $A_t$ may depend on arbitrary past variables. A possible causal graph is shown in Fig. 1. Note that w.l.o.g. we do not consider separate static covariates

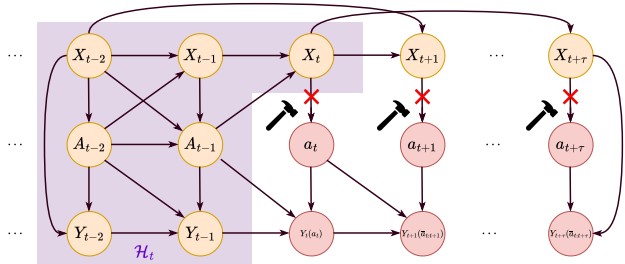

Figure 1: Setting for estimating treatment effects over time.

(e.g., age, gender) as these can be part of $X_t$. We are further given an observational dataset $\mathcal{D} = \left\{ \{x_t^{(i)}, a_t^{(i)}, y_t^{(i)}\}_{t=1}^{T^{(i)}} \right\}_{i=1}^{n}$ consisting of $n \in \mathbb{N}$ patient trajectories sampled i.i.d. from an unknown probability distribution $\mathbb{P}$ of potentially different lengths $T^{(i)}$.

**Causal estimands:** We build upon the potential outcomes framework (Rubin, 1974) for time-varying settings (Robins et al., 2000; Robins & Hernán, 2009) to formally define HTEs. For a time $t$, we consider a fixed sequence of treatment interventions $\bar{a}_{t:t+\tau} = (a_t, \ldots, a_{t+\tau})$ of length $\tau + 1$. We denote the potential outcome $Y_{t+\tau}(\bar{a}_{t:t+\tau})$ as the $\tau$-step-ahead outcome that would be observed when intervening on the treatments $\bar{A}_{t:t+\tau}$ and setting them to $\bar{a}_{t:t+\tau}$. We are then interested in estimating the *conditional average potential outcome (CAPO)*

$$\mu_{\bar{a}}(\bar{h}_t) = \mathbb{E}\left[ Y_{t+\tau}(\bar{a}_{t:t+\tau}) \mid \bar{H}_t = \bar{h}_t \right] \tag{1}$$

and the *conditional average treatment effect (CATE)*

$$\tau_{\bar{a},\bar{b}}(\bar{h}_t) = \mathbb{E}\left[ Y_{t+\tau}(\bar{a}_{t:t+\tau}) - Y_{t+\tau}(\bar{b}_{t:t+\tau}) \mid \bar{H}_t = \bar{h}_t \right], \tag{2}$$

where $\bar{b}_{t:t+\tau}$ denotes another sequence of treatment interventions. Both CAPO (and CATE) are functions that predict (differences in) the $\tau$-step-ahead potential outcomes based on the history at time period $t$.

**Identifiability.** We impose the following assumptions to ensure the identifiability of both CAPO and CATE from observational data (for a proof we refer to Appendix C). These assumptions are standard in the causal inference literature (Bica et al., 2020a; Lim et al., 2018; Melnychuk et al., 2022). Practically, they exclude spillover effects, deterministic treatment assignments, and unobserved confounders (Frauen et al., 2023b).

**Assumption 1** (Identifiability (Robins et al., 2000)). We assume that the following conditions hold for all $\bar{a}_{t:t+\tau}$ and $\bar{h}_t$: (i) *consistency*: $Y_{t+\tau}(\bar{a}_{t:t+\tau}) = Y_{t+\tau}$ whenever $\bar{A}_{t:t+\tau} = \bar{a}_{t:t+\tau}$ ; (ii) *overlap*: $\mathbb{P}(A_t = a_t \mid \bar{H}_t = \bar{h}_t) > 0$ whenever $\mathbb{P}(\bar{H}_t = \bar{h}_t) > 0$ ; and (iii) *sequential ignorability*: $A_t \perp Y_{t+\delta}(a_{t+\delta}) \mid \bar{H}_t = \bar{h}_t$ for all $\delta \in \{t, \ldots, t+\tau\}$.

**Time-varying confounding and adjustments.** A key challenge of the time-varying setting is that the unbiased estimation of CAPO and CATE is non-trivial. Under Assumption 1, both CAPO and CATE are identified and can thus be expressed as a function of the observational data distribution (Robins, 1999; Robins et al., 2000). However, the time-varying nature of the confounding structure gives rise to adjustment mechanisms that are different from the static setting whenever $\tau > 0$. In particular, adjusting for only the history $H_t$ is generally insufficient for unbiased estimation (van der Laan & Gruber, 2012). That is, for $\tau > 0$, it holds that

$$\mathbb{E}\left[Y_{t+\tau}(\bar{a}_{t:t+\tau}) \mid \bar{H}_t = \bar{h}_t\right] \neq \mathbb{E}\left[Y_{t+\tau} \mid \bar{H}_t = \bar{h}_t, \bar{A}_{t:t+\tau} = \bar{a}_{t:t+\tau}\right]. \tag{3}$$

Importantly, Eq. (3) implies that static meta-learners that only condition on the history $\bar{H}_t$ are *biased* in the time-varying setting. The reason is that the history adjustment in Eq. (3) ignores to adjust for the *post-treatment confounders* $X_\ell$ for $\ell > t$, which are not observed during inference time. This phenomenon is also known as *time-varying confounding*[2]. Adjusting for post-treatment confounders requires leveraging custom adjustment formulas for the time-varying settings. This includes iterated conditional expectations (G-computation) (van der Laan & Gruber, 2012) and products of propensity scores (inverse-propensity weighting) (Robins et al., 2000). In the next section, we leverage such time-varying adjustment formulas to construct unbiased meta-learners for estimating both CAPO and CATE.

## 4 META-LEARNERS FOR ESTIMATING HTES OVER TIME

In this section, we propose several meta-learners for estimating CAPO and CATE. The general idea of meta-learners is to first estimate so-called *nuisance functions* that capture important aspects of the observational data distribution (e.g., response functions or propensity scores).

Then, the estimated nuisance functions are used for estimating the quantity of interest (e.g., CAPO or CATE). Both nuisance functions and meta-learners are defined in terms of conditional expectations and probabilities that can be estimated using arbitrary machine learning models.

Our meta-learners can be grouped along two criteria: (a) which *adjustment mechanism* is leveraged; and (b) whether the learner is a *plug-in learner* or a *two-stage learner*. For (a), we leverage different adjustment mechanisms, including history adjustment, regression adjustment (G-computation), propensity adjustment, and dou-

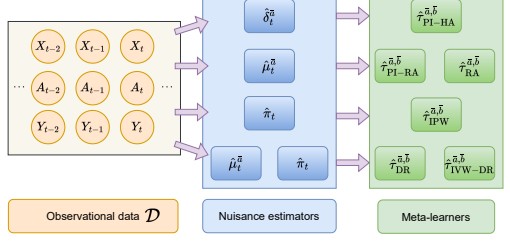

Figure 2: Overview of the different nuisance estimators and meta-learners proposed in this paper.

bly robust adjustment. For (b), the difference between plug-in and two-stage learners is as follows: Plug-in learners "plug" the nuisance functions into the corresponding adjustment formula to estimate the quantity of interest. In contrast, two-stage learners estimate the treatment effect directly by performing a second-stage regression on a pseudo-outcome constructed from the estimated nuisance

---

[2]This is closely related to *runtime confounding* (Coston et al., 2020), where confounders are unavailable during inference time.

functions. Two-stage learners often outperform plug-in learners if the target quantity (e.g., CATE) is of simpler structure and thus easier to estimate than the nuisance parameters involved (Curth et al., 2020; Kennedy, 2023).

In the following, we group our learners according to (a) and propose both plug-in and two-stage learners for different adjustment mechanisms. We refer to Table 5 for a detailed overview and comparison of our meta-learners.

### 4.1 HISTORY ADJUSTMENT

• **PI-HA-learner.** A naïve approach to construct a meta-learner is to use the history adjustment $\delta^{\bar{a}}(\bar{h}_t) = \mathbb{E}\left[Y_{t+\tau} \mid \bar{H}_t = \bar{h}_t, \bar{A}_{t:t+\tau} = \bar{a}_{t:t+\tau}\right]$. As a result, this learner controls *only* for the history $H_t$, while it thus ignores post-treatment confounders. Hence, for some arbitrary estimator $\hat{\delta}^{\bar{a}}(\bar{h}_t)$, we define the *plug-in history-adjustment learner* (PI-HA-learner) for CAPO and CATE as

$$\hat{\mu}^{\bar{a}}_{\text{PI-HA}}(\bar{h}_t) = \hat{\delta}^{\bar{a}}(\bar{h}_t) \quad \text{and} \quad \hat{\tau}^{\bar{a},\bar{b}}_{\text{PI-HA}}(\bar{h}_t) = \hat{\delta}^{\bar{a}}(\bar{h}_t) - \hat{\delta}^{\bar{b}}(\bar{h}_t). \tag{4}$$

Here, the estimator $\hat{\delta}^{\bar{a}}(\bar{h}_t)$ of the nuisance function $\delta^{\bar{a}}(\bar{h}_t)$ is plugged into the formulas for CAPO and CATE, respectively. Note that both $\hat{\tau}^{\bar{a}}_{\text{PI-HA}}(\bar{h}_t)$ and $\hat{\tau}^{\bar{a},\bar{b}}_{\text{PI-HA}}(\bar{h}_t)$ are *biased* because the history adjustment is *not* sufficient for the time-varying setting (see Eq. (3)). The reason why we nevertheless consider the PI-HA-learner is its low variance, which can lead to a relatively good performance in low-sample settings when estimation variance has a larger impact than confounding bias.

The PI-HA-learner can also be interpreted as a plug-in learner from the static setting (i.e., ignoring time) (Curth et al., 2020), where the history $\bar{H}_t$ acts as confounders and the intervention sequence $\bar{A}_{t:t+\tau}$ acts as a discrete treatment. On this basis, one could, in principle, adopt other two-stage learners from the static setting that can handle discrete treatments (Acharki et al., 2023) (e.g., doubly robust versions of history adjustment). Of note, such static two-stage learners suffer from the same bias as the PI-HA-learner as they aim at estimating $\delta^{\bar{a}}(\bar{h}_t)$ but $\underline{\text{not}}$ $\mathbb{E}\left[Y_{t+\tau}(\bar{a}_{t:t+\tau}) \mid \bar{H}_t = \bar{h}_t\right]$.

### 4.2 REGRESSION ADJUSTMENT (G-COMPUTATION)

We now introduce an *unbiased* meta-learner for estimating CAPO and CATE based on regression adjustment, also known as G-computation (Robins, 1999). The nuisance functions used for this are the response functions, defined via

$$\mu^{\bar{a}}_{t+\tau}(\bar{h}_{t+\tau}) = \mathbb{E}\left[Y_{t+\tau} \mid \bar{H}_{t+\tau} = \bar{h}_{t+\tau}, A_{t+\tau} = a_{t+\tau}\right] \tag{5}$$

and iteratively via

$$\mu^{\bar{a}}_{\ell}(\bar{h}_{\ell}) = \mathbb{E}\left[\mu^{\bar{a}}_{\ell+1}(\bar{H}_{\ell+1}) \mid \bar{H}_{\ell} = \bar{h}_{\ell}, A_{\ell} = a_{\ell}\right] \tag{6}$$

for $\ell \in \{t, \dots, t+\tau-1\}$.

• **PI-RA-learner.** Given estimators of the response functions, we introduce the *plug-in regression-adjustment learner* (PI-RA-learner) for CAPO and CATE via

$$\hat{\mu}^{\bar{a}}_{\text{PI-RA}}(\bar{h}_t) = \hat{\mu}^{\bar{a}}_t(\bar{h}_t) \quad \text{and} \quad \hat{\tau}^{\bar{a},\bar{b}}_{\text{PI-RA}}(\bar{h}_t) = \hat{\mu}^{\bar{a}}_t(\bar{h}_t) - \hat{\mu}^{\bar{b}}_t(\bar{h}_t), \tag{7}$$

respectively. The plug-in learner is *unbiased* (see Sec. 5). This is because, under Assumption 1, CAPO can be written as $\tau^{\bar{a}}(\bar{h}_t) = \mu^{\bar{a}}_t(\bar{h}_t)$ (van der Laan & Gruber, 2012; Frauen et al., 2023a). In particular, $\hat{\tau}^{\bar{a}}_{\text{PI-RA}}(\bar{h}_t)$ adjusts for the post-treatment covariates by successively integrating them out through an iterative regression process. This approach is also called *iterative G-computation* and has been leveraged for estimating *average* treatment effects (van der Laan & Gruber, 2012; Frauen et al., 2023a), while we extend the idea to estimating *heterogenous* treatment effects.

• **RA-learner.** We now introduce the *regression-adjustment learner* (RA-learner), which is a two-stage learner for estimating CATE using regression adjustment. The motivation is that, in many applications, the CATE $\tau_{\bar{a},\bar{b}}(\bar{h}_t)$ is of simpler structure than the CAPO $\tau_{\bar{a}}(\bar{h}_t)$ due to cancellation effects (Kennedy, 2023). Hence, two-stage learners that target CATE directly may outperform the plug-in learner in such scenarios. Given estimators $\hat{\mu}^{\bar{a}}_{\ell}(\bar{h}_{\ell})$ of the response functions, we define the pseudo-outcomes

$$\hat{Y}^{\bar{a},\bar{b}}_{\text{RA}} = \mathbb{1}\{A_t = a_t\}\left(\hat{\mu}^{\bar{a}}_{t+1}(\bar{H}_{t+1}) - \hat{\mu}^{\bar{b}}_t(\bar{H}_t)\right) + \mathbb{1}\{A_t = b_t\}\left(\hat{\mu}^{\bar{a}}_t(\bar{H}_t) - \hat{\mu}^{\bar{b}}_{t+1}(\bar{H}_{t+1})\right)$$

$$+ \mathbb{1}\{A_t \neq a_t\}\mathbb{1}\{A_t \neq b_t\}\left(\hat{\mu}^{\bar{a}}_t(\bar{h}_t) - \hat{\mu}^{\bar{b}}_t(\bar{h}_t)\right). \tag{8}$$

We define the RA-learner via the second-stage regression $\hat{\tau}_{\mathrm{RA}}^{\bar{a},\bar{b}}(\bar{h}_t) = \hat{\mathbb{E}}\left[\hat{Y}_{\mathrm{RA}}^{\bar{a},\bar{b}} \mid \bar{H}_t = \bar{h}_t\right]$.

## 4.3 PROPENSITY ADJUSTMENT

Another major type of adjustment is propensity adjustment (Curth et al., 2020). For this, let us define propensity scores in the time-varying setting via

$$\pi_\ell(a_\ell \mid \bar{h}_\ell) = \mathbb{P}(A_\ell = a_\ell \mid \bar{H}_\ell = \bar{h}_\ell) \tag{9}$$

for $\ell \in \{t, \ldots, t + \tau\}$. Each propensity score $\pi_\ell(a_\ell \mid \bar{h}_\ell)$ characterizes the treatment assignment mechanism at time $\ell$ based on the history $\bar{h}_\ell$.

• **IPW-learner.** We now propose the *inverse-propensity-weighted learners* (IPW-learner), which is a two-stage learner for estimating CAPO and CATE based on propensity adjustment. Using estimators $\hat{\pi}_\ell(a_\ell \mid \bar{h}_\ell)$ of the propensity scores, we define the pseudo-outcomes

$$\hat{Y}_{\mathrm{IPW}}^{\bar{a}} = \prod_{\ell=t}^{t+\tau} \frac{\mathbb{1}\{A_\ell = a_\ell\}}{\hat{\pi}_\ell(a_\ell \mid \bar{H}_\ell)} Y_{t+\tau} \quad \text{and} \quad \hat{Y}_{\mathrm{IPW}}^{\bar{a},\bar{b}} = \hat{Y}_{\mathrm{IPW}}^{\bar{a}} - \hat{Y}_{\mathrm{IPW}}^{\bar{b}}. \tag{10}$$

Our IPW-learner is a two-stage learner defined via the second-stage regressions

$$\hat{\mu}_{\mathrm{IPW}}^{\bar{a}}(\bar{h}_t) = \hat{\mathbb{E}}\left[\hat{Y}_{\mathrm{IPW}}^{\bar{a}} \mid \bar{H}_t = \bar{h}_t\right] \quad \text{and} \quad \hat{\tau}_{\mathrm{IPW}}^{\bar{a},\bar{b}}(\bar{h}_t) = \hat{\mathbb{E}}\left[\hat{Y}_{\mathrm{IPW}}^{\bar{a},\bar{b}} \mid \bar{H}_t = \bar{h}_t\right]. \tag{11}$$

Inverse-propensity weighting in the time-varying setting has originally been proposed for estimating *average* treatment effects (e.g., via marginal structural models (Robins et al., 2000)), however, not explicitly for *heterogeneous* treatment effects.

## 4.4 DOUBLY ROBUST ADJUSTMENT

The above learners (i.e., the PI-HA-, PI-RA-, RA-, and IPW-leaners) have a shortcoming in that they all utilize estimators of only some of the possible nuisance functions. While the PI-HA-, PI-RA-, and RA-learners leverage *only* the response function estimators (or parts thereof), the IPW-learner leverages *only* the propensity score estimators. This motivates us to construct meta-learners that leverage *both* nuisance estimators and thus all components informative of the data-generating process.

• **DR-learner.** We propose the *doubly robust learner* (DR-learner), which is a two-stage learner that can be used for estimating both CAPO and CATE. For this, we define the pseudo-outcomes

$$\hat{Y}_{\mathrm{DR}}^{\bar{a}} = \hat{Y}_{\mathrm{IPW}}^{\bar{a}} + \sum_{k=t}^{t+\tau} \hat{\mu}_k^{\bar{a}}\left(\bar{H}_k\right)\left(1 - \frac{\mathbb{1}\{A_k = a_k\}}{\hat{\pi}_k(a_k \mid \bar{H}_k)}\right) \prod_{\ell=t}^{k-1} \frac{\mathbb{1}\{A_\ell = a_\ell\}}{\hat{\pi}_\ell(a_\ell \mid \bar{H}_\ell)} \quad \text{and} \quad \hat{Y}_{\mathrm{DR}}^{\bar{a},\bar{b}} = \hat{Y}_{\mathrm{DR}}^{\bar{a}} - \hat{Y}_{\mathrm{DR}}^{\bar{b}}. \tag{12}$$

The pseudo-outcomes $\hat{Y}_{\mathrm{DR}}^{\bar{a}}$ and $\hat{Y}_{\mathrm{DR}}^{\bar{a},b}$ are estimators of the (uncentered) efficient influence function of CAPO and CATE, respectively (van der Laan & Gruber, 2012). Similar to the static setting (Kennedy, 2023), we introduce our DR-learner for the time-varying setting via second-stage regressions based on the efficient influence functions. That is, we define

$$\hat{\mu}_{\mathrm{DR}}^{\bar{a}}(\bar{h}_t) = \hat{\mathbb{E}}\left[\hat{Y}_{\mathrm{DR}}^{\bar{a}} \mid \bar{H}_t = \bar{h}_t\right] \quad \text{and} \quad \hat{\tau}_{\mathrm{DR}}^{\bar{a},\bar{b}}(\bar{h}_t) = \hat{\mathbb{E}}\left[\hat{Y}_{\mathrm{DR}}^{\bar{a},\bar{b}} \mid \bar{H}_t = \bar{h}_t\right]. \tag{13}$$

Due to the use of efficient influence functions, the DR-learner comes with theoretical properties such as double robustness (see Sec. 5).

## 5 THEORETICAL ANALYSIS

In this section, we add a theoretical analysis for the PI-HA-, PI-RA-, RA-, IPW-, and DR-learners. For this, we employ mathematical tools from Kennedy (2023) and derive *asymptotic bounds* on the point-wise risk of the corresponding learners. We start by imposing the following assumptions.

**Assumption 2** (Boundedness). The response functions $\mu_\ell^{\bar{a}}(\cdot)$, the propensity scores $\pi_\ell(a_\ell \mid \cdot)$, and its estimated versions $\hat{\mu}_\ell^{\bar{a}}(\cdot)$ and $\hat{\pi}_\ell(a_\ell \mid \cdot)$ are bounded functions for all interventions $\bar{a}$.

**Assumption 3** (Estimation). The regression estimators $\hat{\mathbb{E}}_n$ admit rates $r_{\mu_\ell}^{\bar{a}}(n)$ for the response functions $\hat{\mu}_\ell^{\bar{a}}(\cdot)$, $r_{\text{PI-HA}}^{\bar{a}}(n)$ for the history adjustments $\hat{\delta}^{\bar{a}}(\cdot)$, and $r_{\text{PO}}^{\bar{a},\bar{a}}(n)$ for the second-stage estimators $\hat{\mathbb{E}}$ (see Appendix D for a mathematical definition and examples). We also impose the stability assumptions from Kennedy (2023) on all regression estimators $\hat{\mathbb{E}}_n$ (see Appendix C). Finally, the propensity score estimators $\hat{\pi}_\ell(a_\ell \mid \cdot)$ admit rates $r_{\pi_\ell}^{\bar{a}}(n)$.

**Assumption 4** (Sample splitting). For each learner, the observational dataset $\mathcal{D}$ admits a partition $\mathcal{D} = \mathcal{D}_t^{\hat{\mu}} \cup \cdots \cup \mathcal{D}_{t+\tau}^{\hat{\mu}} \cup \mathcal{D}^{\text{PO}}$, so that each response function estimator $\hat{\mu}_\ell^{\bar{a}}$ is fitted on $\mathcal{D}_\ell^{\hat{\mu}}$, all propensity score estimators $\hat{\pi}_\ell$ are fitted on $\mathcal{D} \setminus \mathcal{D}^{\text{PO}}$, and any second-stage regressor $\hat{\mathbb{E}}$ is fitted on $\mathcal{D}^{\text{PO}}$.

Boundedness assumptions such as Assumption 2 are standard in the causal inference literature to provide theoretical guarantees for treatment effect learners (Curth et al., 2020; Foster & Syrgkanis, 2023; Frauen & Feuerriegel, 2023; Kennedy, 2023). Assumption 3 on the estimators is in line with Kennedy (2023). The rates $r_{\mu_\ell}^{\bar{a}}(n)$, $r_{\pi_\ell}^{\bar{a}}(n)$, $r_{\text{PI-HA}}^{\bar{a}}(n)$, and $r_{\text{PO}}^{\bar{a}}(n)$ capture assumptions on the *complexity* of the corresponding estimands, i.e., nuisance parameters and CATE. The assumption can be instantiated via, e.g., enforcing smoothness or sparsity on the underlying model (Curth et al., 2020; Kennedy, 2023). Finally, Assumption 4 (i.e., sample splitting) is standard in the literature on meta-learners for causal inference (Chernozhukov et al., 2018; Foster & Syrgkanis, 2023; Kennedy, 2023). While sample splitting is required for the validity of our theoretical results, we note that, in practice, all of our learners can also be used without sample splitting. Avoiding the use of sample splitting can result in superior finite-sample performance as more data is used to estimate nuisance parameters and second-stage regressions (Curth et al., 2020). In our experiments, we follow established literature (Curth & van der Schaar, 2021) and refrain from using sample splitting.

The following result establishes asymptotic rates for the risk our meta-learners for CATE under treatment interventions $\bar{a}$ and $\bar{b}$. Rates for CAPO can be obtained by removing all terms that include $\bar{b}$. We write $g(n) \lesssim f(n)$ whenever there exist $C, n_0 > 0$, so that $g(n) \leq C f(n)$ for all $n > n_0$.

**Theorem 1.** *Under Assumptions 1–4, we obtain the following rates:*

$$\mathbb{E}[(\hat{\tau}_{\text{PI-HA}}^{\bar{a},\bar{b}}(\bar{h}_t) - \tau_{\bar{a},\bar{b}}(\bar{h}_t))^2] \lesssim r_{\text{PI-HA}}^{\bar{a}}(n) + r_{\text{PI-HA}}^{\bar{b}}(n) + \text{bias}_{\bar{a},\bar{b}}^2 \tag{14}$$

$$\mathbb{E}[(\hat{\tau}_{\text{PI-RA}}^{\bar{a},\bar{b}}(\bar{h}_t) - \tau_{\bar{a},\bar{b}}(\bar{h}_t))^2] \lesssim \sum_{\ell=t}^{t+\tau} r_{\mu_\ell}^{\bar{a}}(n) + r_{\mu_\ell}^{\bar{b}}(n) \tag{15}$$

$$\mathbb{E}[(\hat{\tau}_{\text{RA}}^{\bar{a},\bar{b}}(\bar{h}_t) - \tau_{\bar{a},\bar{b}}(\bar{h}_t))^2] \lesssim r_{\text{PO}}^{\bar{a},\bar{b}}(n) + \sum_{\ell=t}^{t+\tau} r_{\mu_\ell}^{\bar{a}}(n) + r_{\mu_\ell}^{\bar{b}}(n) \tag{16}$$

$$\mathbb{E}[(\hat{\tau}_{\text{IPW}}^{\bar{a},\bar{b}}(\bar{h}_t) - \tau_{\bar{a},\bar{b}}(\bar{h}_t))^2] \lesssim r_{\text{PO}}^{\bar{a},\bar{b}}(n) + \sum_{\ell=t}^{t+\tau} r_{\pi_\ell}^{\bar{a}}(n) + r_{\pi_\ell}^{\bar{b}}(n) \tag{17}$$

$$\mathbb{E}[(\hat{\tau}_{\text{DR}}^{\bar{a},\bar{b}}(\bar{h}_t) - \tau_{\bar{a},\bar{b}}(\bar{h}_t))^2] \lesssim r_{\text{PO}}^{\bar{a},\bar{b}}(n) + \sum_{\ell=t}^{t+\tau} \left( r_{\pi_\ell}^{\bar{a}}(n) \sum_{k=\ell}^{t+\tau} r_{\mu_k}^{\bar{a}}(n) + r_{\pi_\ell}^{\bar{b}}(n) \sum_{k=\ell}^{t+\tau} r_{\mu_k}^{\bar{b}}(n) \right), \tag{18}$$

$$\text{bias}_{\bar{a},\bar{b}} = \mathbb{E}\left[Y_{t+\tau} \mid \bar{H}_t = \bar{h}_t, \bar{A}_{t:t+\tau} = \bar{a}_{t:t+\tau}\right] - \mathbb{E}\left[Y_{t+\tau} \mid \bar{H}_t = \bar{h}_t, \bar{A}_{t:t+\tau} = \bar{b}_{t:t+\tau}\right] - \tau_{\bar{a},\bar{b}}(\bar{h}_t).$$

*Proof.* See Appendix C. □

For $\tau = 0$, the rates of all learners coincide with the corresponding rates in the static setting (Curth & van der Schaar, 2021; Kennedy, 2023).

**Interpretation of Theorem 1.** Theorem 1 provides us with the following insights, assuming the rate $r_{\text{PO}}^{\bar{a},\bar{b}}(n)$ vanishes faster than the nuisance rates (e.g., when CATE is of simpler structure): (i) The PI-HA-learner is biased, even asymptotically, when $n \to \infty$. In contrast, all other learners are unbiased (consistent) whenever $n \to \infty$, assuming the involved rates vanish. (ii) The PI-RA- and RA-learner are asymptotically equivalent. (iii) The rate of each meta-learner depends on the rate of the corresponding nuisance parameters. Hence, the PI-RA- and RA-learner will generally converge faster whenever the response functions are easier to estimate, while the IPW-learner will converge

faster whenever the propensity scores are easier to estimate. (iv) The DR-learner admits a *doubly robust rate*. That is, the DR-learner will converge at a fast rate if each $\ell$, either the propensity score $\pi_\ell(a_\ell \mid \cdot)$ or *all* response functions $\mu_k^{\bar{a}}(\cdot)$ for $k \geq \ell$ allow for fast rates (for both $\bar{a}$ and $\bar{b}$).

## 6 STABILIZING THE DR-LOSS VIA INVERSE VARIANCE WEIGHTING

A drawback of the DR-learner is the division by propensity scores in Eq. (12), which can render the second-stage regression unstable whenever the propensity score estimates $\hat{\pi}_\ell(a_\ell \mid \bar{h}_\ell)$ are small. While this is a well-known issue in the static setting (Kennedy et al., 2024; Morzywolek et al., 2023), it is especially problematic in the time-varying setting where the DR-learner uses *products* of inverse-propensity scores (Bica et al., 2020a). Hence, we now propose a method for reducing the variance of the DR-learner.

Our idea is motivated by a recent work from Fisher (2024) who proposed to stabilize the DR-learner from the static setting by using so-called inverse-variance weights (IVWs). This is analogous to the R-learner from the static setting, which can be written as an inverse-variance weighted version of another learner (the U-learner (Künzel et al., 2019)). We now extend the aforementioned results from the static to the *time-varying* setting. The following theorem derives the inverse-variance weights for the DR-learner in the time-varying setting.

**Theorem 2.** *We define the weights*

$$V_{\mathrm{DR}}^{\bar{a}} = \sum_{k=t}^{t+\tau} \prod_{\ell=t}^{k} \frac{\mathbb{1}\{A_\ell = a_\ell\}}{\pi_\ell^2(a_\ell \mid \bar{H}_\ell)} \quad \text{and} \quad V_{\mathrm{DR}}^{\bar{a},\bar{b}} = V_{\mathrm{DR}}^{\bar{a}} + V_{\mathrm{DR}}^{\bar{b}}. \tag{19}$$

*If it holds that* $\mathrm{Var}\left(\mu_{k+1}^{\bar{b}}\left(\bar{H}_{k+1}\right) \mid \bar{H}_k, A_k\right) = \sigma^2$ *for all* $k \in \{t, \ldots, t+\tau\}$, *we obtain*

$$\mathrm{Var}(Y_{\mathrm{DR}}^{\bar{a}} \mid \bar{H}_t = \bar{h}_t) = \sigma^2 \mathbb{E}\left[V_{\mathrm{DR}}^{\bar{a}} \mid \bar{H}_t = \bar{h}_t\right] \text{ and } \mathrm{Var}(Y_{\mathrm{DR}}^{\bar{a},\bar{b}} \mid \bar{H}_t = \bar{h}_t) = \sigma^2 \mathbb{E}\left[V_{\mathrm{DR}}^{\bar{a},\bar{b}} \mid \bar{H}_t = \bar{h}_t\right].$$

*Proof.* See Appendix C. □

● **IVW-DR-learner.** Motivated by Theorem 2, we propose an *inverse-variance-weighted doubly-robust learner* (IVW-DR-learner) for estimating CAPO and CATE. In contrast to our DR-learner from above, the IPW-DR-learner is not defined via a second-stage regression on a pseudo-outcome. Instead, we introduce it as a minimizer of a loss weighted with stabilized inverse-variance weights. Using estimators $\hat{W}_{\mathrm{DR}}^{\bar{a}} = \hat{\mathbb{E}}\left[\hat{V}_{\mathrm{DR}}^{\bar{a}} \mid \bar{H}_t\right]$ and $\hat{W}_{\mathrm{DR}}^{\bar{a},\bar{b}} = \hat{\mathbb{E}}\left[\hat{V}_{\mathrm{DR}}^{\bar{a},\bar{b}} \mid \bar{H}_t\right]$, we now define the IPW-DR-learner as

$$\hat{\mu}_{\mathrm{IVW}}^{\bar{a}} = \arg\min_{\tau'} \hat{\mathbb{E}}\left[\frac{\hat{W}_{\mathrm{DR}}^{\bar{a}-1}}{\hat{\mathbb{E}}[\hat{W}_{\mathrm{DR}}^{\bar{a}-1}]}\left(\hat{Y}_{\mathrm{DR}}^{\bar{a}} - \tau'(\bar{H}_t)\right)^2\right] \text{ and } \hat{\tau}_{\mathrm{IVW}}^{\bar{a},\bar{b}} = \arg\min_{\tau'} \hat{\mathbb{E}}\left[\frac{\hat{W}_{\mathrm{DR}}^{\bar{a}\,\bar{b}-1}}{\hat{\mathbb{E}}[\hat{W}_{\mathrm{DR}}^{\bar{a}\,\bar{b}-1}]}\left(\hat{Y}_{\mathrm{DR}}^{\bar{a},\bar{b}} - \tau'(\bar{H}_t)\right)^2\right]. \tag{20}$$

We emphasize that the assumption of constant variance in Theorem 2 is in line with the assumption from (Fisher, 2024) to interpret the weights of the R-learner (Nie & Wager, 2021) in the static setting as inverse-variance weights. Even if the assumption is violated, the weights may still stabilize the doubly robust loss despite not being interpretable as inverse-variance weights.

To gain insights into the weighting mechanism of the IVW-DR-learner, we make a comparison to the static setting where $\tau = 0$. Here, we obtain $\mathbb{E}[V_{\mathrm{DR}}^{\bar{a},\bar{b}} \mid \bar{H}_t = \bar{h}_t] = \pi_t(a_\ell \mid \bar{h}_t)^{-1} + \pi_t(b_\ell \mid \bar{h}_t)^{-1} = \frac{\pi_t(a_\ell|\bar{h}_t)+\pi_t(b_\ell|\bar{h}_t)}{\pi_t(a_\ell|\bar{h}_t)\pi_t(b_\ell|\bar{h}_t)}$ for $\tau = 0$. Hence, for binary treatments, which imply $\pi_t(a_\ell \mid \bar{h}_t)+\pi_t(b_\ell \mid \bar{h}_t) = 1$, the inverse variance weights reduce to the product of propensities $\pi_t(a_\ell \mid \bar{h}_t)\pi_t(b_\ell \mid \bar{h}_t)$. These weights reduce the influence of samples with low overlap, which may lead to instabilities due to a lack of data. This corresponds to the DR-learner with inverse-variance weights in the static setting. In particular, our IVWs are a generalization to *both* time-varying and multiple treatments.

## 7 EXPERIMENTS

In this section, we compare our proposed meta-learners empirically. The purpose of our experiments is two-fold: (i) we *verify the insights from our theoretical results* (Sec. 5); and (ii) we show that *our*

*IVW-DR-learner improves over the DR-learner* and other meta-learners, specifically in low-overlap regimes and for long time horizons $\tau$. Recall that all our results are model-agnostic and therefore hold for arbitrary machine learning instantiations. To ensure a fair comparison, we estimate all nuisance functions and instantiate all meta-learners using the same transformer-based architecture. In particular, *we refrain from comparing with model-based methods* as these can be viewed as instantiations of certain meta-learners (see Table 5) using specific models. Of note, this is consistent with established literature on meta-learners in the static setting (Curth & van der Schaar, 2021). To empirically verify robustness with respect to model choice, we performed additional experiments using LSTM-based instantiations of our meta-learners in Appendix H. We also provide additional results in Appendix I

**Neural instantiations.** In our experiments, we use transformer-based instantiations of our meta-learners. That is, we estimate both our nuisance function estimators and second-stage regressions using deep neural networks, specifically transformer architectures. Transformers (Ashish Vaswani et al., 2017) are effective at learning representations that achieve impressive performance on sequential learning problems, including causal inference tasks (Melnychuk et al., 2022). However, we emphasize that our implementation should be seen as a "proof-of-concept". This means that we do not claim that the architecture of our transformer instantiations is optimized for task-specific performance. Rather, practitioners will need to determine the preferred model architecture based on various factors such as sample size or data dimensionality (Curth & van der Schaar, 2021). Nevertheless, this also pinpoints a clear advantage of our meta-learners: they can be used with arbitrary machine learning models and allow various other extensions (e.g., balancing losses; see Appendix F for a discussion). In Appendix H, we provide additional results using LSTM-based instantiations.

We build upon an encoder-only transformer from Ashish Vaswani et al. (2017) with a causal mask to avoid look-ahead bias and non-trainable positional encodings, combined with a feed-forward neural network with linear or softmax activation. We perform

Table 2: Results for $\mathcal{D}_1$.

|  | $\tau = 0$ | $\tau = 1$ | $\tau = 2$ | $\tau = 3$ | $\tau = 4$ |
|---|---|---|---|---|---|
| PI-HA-learner | $1.57 \pm 0.38$ | $10.92 \pm 1.24$ | $16.00 \pm 2.01$ | $14.45 \pm 5.30$ | $7.56 \pm 2.42$ |
| PI-RA-learner | $1.60 \pm 0.36$ | $2.57 \pm 0.53$ | $2.62 \pm 0.57$ | $2.60 \pm 0.74$ | $1.72 \pm 0.25$ |
| RA-learner | $0.95 \pm 0.46$ | $1.32 \pm 0.59$ | $\mathbf{2.13 \pm 0.83}$ | $\mathbf{2.29 \pm 0.80}$ | $1.39 \pm 0.29$ |
| IPW-learner | $0.60 \pm 0.28$ | $1.43 \pm 0.81$ | $14.77 \pm 6.78$ | $8.25 \pm 4.46$ | $5.03 \pm 3.03$ |
| DR-learner | $0.65 \pm 0.35$ | $1.39 \pm 0.70$ | $14.60 \pm 6.50$ | $6.82 \pm 2.45$ | $5.70 \pm 3.84$ |
| IVW-DR-learner | $\mathbf{0.57 \pm 0.14}$ | $\mathbf{1.29 \pm 0.23}$ | $2.40 \pm 0.43$ | $2.36 \pm 1.12$ | $\mathbf{1.28 \pm 0.36}$ |

Reported: Average RMSE $\pm$ standard deviation ($\times 10$) over
5 random seeds (best in bold).

training using the Adam optimizer (Kingma & Ba, 2015). Further details regarding architecture, training, hyperparameters, and runtime are in Appendix E.

**Simulated datasets.** We simulate three datasets $\mathcal{D}_j$ with $j \in \{1, 2, 3\}$ from different data-generating processes. Of note, the use of simulated data is standard in the causal inference literature (Lim et al., 2018; Bica et al., 2020a; Curth et al., 2020; Frauen & Feuerriegel, 2023) and enables measurement of the performance of causal inference methods due to known ground-truth. We report full details regarding the generation of all datasets $\mathcal{D}_j$ in Appendix G.

Table 3: Results for $\mathcal{D}_2$.

|  | $\tau = 0$ | $\tau = 1$ | $\tau = 2$ | $\tau = 3$ | $\tau = 4$ |
|---|---|---|---|---|---|
| PI-HA-learner | $2.52 \pm 0.56$ | $1.99 \pm 0.44$ | $2.65 \pm 0.34$ | $2.26 \pm 0.52$ | $3.18 \pm 0.39$ |
| PI-RA-learner | $2.33 \pm 0.30$ | $1.37 \pm 0.06$ | $1.08 \pm 0.30$ | $1.00 \pm 0.39$ | $0.63 \pm 0.39$ |
| RA-learner | $0.51 \pm 0.22$ | $0.53 \pm 0.24$ | $0.75 \pm 0.30$ | $0.89 \pm 0.43$ | $0.56 \pm 0.40$ |
| IPW-learner | $0.20 \pm 0.07$ | $0.42 \pm 0.13$ | $0.58 \pm 0.16$ | $1.46 \pm 0.87$ | $2.62 \pm 0.34$ |
| DR-learner | $0.14 \pm 0.07$ | $0.42 \pm 0.32$ | $\mathbf{0.52 \pm 0.09}$ | $1.84 \pm 0.89$ | $2.69 \pm 0.71$ |
| IVW-DR-learner | $\mathbf{0.11 \pm 0.07}$ | $\mathbf{0.32 \pm 0.31}$ | $0.57 \pm 0.24$ | $\mathbf{0.80 \pm 0.46}$ | $\mathbf{0.50 \pm 0.35}$ |

Reported: average RMSE $\pm$ standard deviation ($\times 10$) over
5 random seeds (best in bold).

At a high level, the datasets satisfy the following properties: (i) All datasets include time-varying confounders $X_t$, which should render the PI-HA-learner biased. (ii) The treatment effect mechanisms of all datasets are "simpler" than the response function mechanisms, which should give two-stage meta-learners an advantage over plug-in meta-learners (see Sec. 5). (iii) The treatment assignment mechanism of $\mathcal{D}_1$ ($\mathcal{D}_2$) is more (less) "complex" than the response function mechanisms, which should give learners based on regression adjustment an advantage (disadvantage) over learners based on propensity adjustment (see Sec. 5). (iv) Both $\mathcal{D}_1$ and $\mathcal{D}_3$ contain observations with low overlap (i.e., large or small propensity scores), which should increase the variance of the IPW- and the DR-learner and increase the relative performance of the IVW-DR-learner. Hence, we now verify whether we can confirm these properties in our numerical experiments.

**Results for $\mathcal{D}_1$ and $\mathcal{D}_2$.** The results for $\mathcal{D}_1$ and $\mathcal{D}_2$ are reported in Table 2 and Table 3, respectively. Here, we report the RMSE of the estimated CATEs for $\tau \in \{0, 1, 2, 3, 4\}$ using different interventions $\bar{a}_\tau$ and $\bar{b}_\tau$ (see Appendix G for details). In line with the properties of the datasets, we observe the following: (i) For $\tau > 0$, the PI-HA achieves the worst performance as compared to the other meta-learners, indicating a bias due to runtime confounding. (ii) The two-stage learners (RA, IPW, DR, and IVW-DR) outperform the plugin learners (PI-HA and PI-RA) on both datasets. (iii) The

RA-learner outperforms the IPW-learner on $\mathcal{D}_1$ (except for $\tau = 0$), while the IPW-learner is better on $\mathcal{D}_2$. In particular, the IPW-learner performs better for smaller prediction horizons $\tau$, which is likely due to increasing variance when dividing by products of propensity scores. The DR- and IVW-DR learners tend to perform well on both datasets, which confirms the doubly robust convergence rate from Theorem 1. (iv) The IPW- and DR-learners are unstable on $\mathcal{D}_1$ for $\tau = 2$, which is likely due to low overlap and divisions by products of propensity scores. Notably, the IVW-DR-learner improves over both by a large margin, indicating a stabilizing effect of the inverse variance weights on the DR-loss. The performance gain is particularly noticeable for larger $\tau$ where the overlap is inherently smaller. In sum, *our empirical findings confirm our theoretical results and the effectiveness of our IVW-DR-learner in scenarios with low overlap and long time horizons.*

**Sensitivity to low overlap ($\mathcal{D}_3$).** We further analyze the sensitivity of the DR-learner to low overlap and compare it with our IVW-DR-learner. To do so, we report the RMSE (for $\tau = 1$) for both learners on $\mathcal{D}_3$ across different levels of overlap (quantified via a parameter $\gamma$; see Appendix G for details). The results are shown in Fig 3. As expected, the DR-learner becomes unstable for lower levels of overlap (i.e., larger $\gamma$). In contrast, the IVW-DR-learner remains stable and outperforms the DR-learner by a large margin for small overlap. *This confirms the effectiveness of our inverse variance weights (Theorem 2) to stabilize the DR-loss, which is crucial in the time-varying setting that comes with inherently low overlap.*

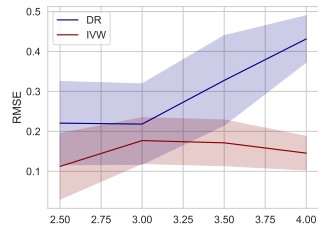

Figure 3: RMSE of DR- and IVW-DR-learner for different levels of overlap averaged over 5 random seeds. Larger $\gamma$ implies lower overlap.

**Real-world dataset.** We sample $n = 3000$ patient trajectories electronic health records over up to $T = 10$ time points from the MIMIC III dataset (Johnson et al., 2016). Similar to Melnychuk et al. (2022), we include 25 time-varying covariates, gender and age as static covariates, ventilation as a binary treatment, and blood pressure as time-varying outcome (standardized with a mean of $61.09$ and a standard deviation of $14.48$). We train our meta-learners for CAPO and prediction horizons $\tau \in \{0, 1, 2, 3\}$ and for the treatment intervention that gives treatment at every time step. We use the factual RMSE (i.e., RMSE between model predictions and observed outcomes whenever observed treatments coincide with the intervention sequence) as a proxy for performance on real-world data. However, we note that a reliable evaluation of causal models on real-world data is notoriously hard due to the fundamental problem of causal inference (Shalit et al., 2017; Curth & van der Schaar, 2021). Hence, our results should be seen as indications and interpreted with care.

Table 4: Results for real-world data.

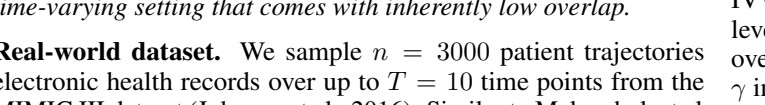

| | $\tau = 0$ | $\tau = 1$ | $\tau = 2$ | $\tau = 3$ |
|---|---|---|---|---|
| PI-HA-learner | **0.36 ± 0.02** | 0.66 ± 0.02 | 0.73 ± 0.03 | 0.79 ± 0.02 |
| PI-RA-learner | **0.36 ± 0.02** | 0.66 ± 0.03 | 0.72 ± 0.02 | 0.78 ± 0.04 |
| IPW-learner | 0.53 ± 0.10 | 0.71 ± 0.04 | 0.78 ± 0.08 | 2.03 ± 2.03 |
| DR-learner | 0.52 ± 0.09 | 0.72 ± 0.03 | 0.97 ± 0.43 | 1.00 ± 0.43 |
| IVW-DR-learner | 0.39 ± 0.03 | **0.62 ± 0.01** | **0.67 ± 0.01** | **0.71 ± 0.02** |

Reported: Average factual RMSE ± standard deviation over 5 random seeds.

**Results:** Interestingly, both the PI-HA and the PI-RA learner perform similarly well, even though the PI-HA learner suffers from bias due to time-varying confounding (Theorem 1). We suspect that this is due to a larger finite-sample variance of the PI-RA learner, as it requires iterative fitting $\tau$ models instead of only one. Both the IPW and DR-learner perform relatively badly and become unstable for larger time horizons. This is likely due to limited overlap which leads to divisions by small (products) of propensities. In contrast, our IVW-DR-learner achieves the best overall performance. *This confirms the effectiveness of our inverse variance weights on real-world data.*

**Discussion.** In this paper, we proposed model-agnostic meta-learners that can be used for estimating CAPO or CATE over time using arbitrary machine learning models. **Limitations.** In this paper, we studied the theoretical foundations of meta-learners and provided practical insights. However, we did not study specific model architectures as this generally depends on the complexity of the data-generating process, overlap, and sample size. **Future directions.** We believe the development of state-of-the-art model architectures that can be used as backbones for our meta-learners for specific applications to be a valuable direction for future research. Furthermore, future research may consider extending our meta-learners to the continuous time setting. **Broader impact.** Our meta-learners have the potential to improve various fields in which personalized sequential decision-making is of importance. However, careful implementation is needed to ensure robustness in practice.

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

# A   COMPARISON OF META-LEARNERS FOR ESTIMATING HTES OVER TIME

**Comparison of different meta-learners.** Table 5 contains an overview of both model-agnostic and model-based learners for estimating HTEs over time. Aside from our meta-learners, the only other model-agnostic learner is the R-learner from (Lewis & Syrgkanis, 2021), which leverages a Neyman-orthogonal loss but imposes parametric assumptions based on structural nested mean models (SNMMs; see Appendix B) (Robins, 1994). In contrast, our DR- and IVW-DR-learners are the only nonparametric learners that leverage efficient influence functions.

**Comparison with model-based learners.** Existing model-based learners for estimating CAPO are instantiations of either (i) the PI-HA-, (ii) the PI-RA, or (iii) the IPW-learner (see Table 5, right column). For (i), existing model-based learners that build upon history adjustments are CRN (Bica et al., 2020a) and the Causal Transformer (CT) (Melnychuk et al., 2022). For (ii), G-Net (Li et al., 2021) leverages a version of G-computation and can thus be considered an instantiation of the PI-RA learner. For (iii), RMSNs (Lim et al., 2018) are a model-based learner using inverse-propensity weighting. To the best of our knowledge, no work has considered doubly robust adjustments for estimating CAPO or CATE in the time-varying setting, which is a novelty of our work.

Table 5: Overview of meta-learners for estimating heterogeneous treatment effects over (discrete) time.

| Adjustment type | Meta-learners | Nonparametric | Unbiased | CAPO | CATE | Runtime**** | IVWs | Instantiations |
|---|---|---|---|---|---|---|---|---|
| History adjustment | PI-HA-learner | ✓ | ✗ | ✓ | $(✓)^*$ | $r$ | ✗ | CRN (Bica et al., 2020a), CT (Melnychuk et al., 2022) |
| Regression adjustment | PI-RA-learner | ✓ | ✓ | ✓ | $(✓)^*$ | $\tau \cdot r$ | ✗ | G-Net (Li et al., 2021), GT (Hess et al., 2024a) |
|  | RA-learner | ✓ | ✓ | ✗ | ✓ | $r \cdot (\tau + 1)$ | ✗ | — |
| Propensity adjustment | IPW-learner | ✓ | ✓ | ✓ | ✓ | $2 \cdot r$ | ✗ | RMSNs (Lim et al., 2018) |
| Doubly robust adjustment | DR-learner | ✓ | ✓ | ✓ | ✓ | $r \cdot (\tau + 2)$ | ✗ | — |
|  | IVW-DR-learner | ✓ | ✓ | ✓ | ✓ | $r \cdot (\tau + 2)$ | ✓ | |
| SNMM | R-learner (Lewis & Syrgkanis, 2021) | $(✗)^{**}$ | $(✓)^{***}$ | ✓ | ✓ | $r \cdot (2\tau + (\tau^2 + \tau)/2)$ | ✗ | — |

* Learners suffer from plug-in bias.   ** Except when $\tau = 0$ and the treatment is binary.   *** If parametric assumptions hold.   **** $r$ denoting the runtime of nuisance models

## B  EXTENDED RELATED WORK

### B.1  MODEL-AGNOSTIC METHODS FOR ESTIMATING AVERAGE TREATMENT EFFECTS OVER TIME

Model-agnostic estimators for the time-varying setting have been proposed for *average* treatment effects. Here, prominent estimators build upon different time-varying adjustment mechanisms, including regression-adjustment (G-computation) (Robins, 1999), propensity adjustment (Robins et al., 2000), and doubly robust adjustment (van der Laan & Gruber, 2012; Chernozhukov et al., 2023). Furthermore, additional estimators have been proposed that impose parametric assumptions on the data-generating process, including estimators based on marginal structural models (MSMs (Robins et al., 2000)) and structural nested mean models (SNMMs (Robins, 1994)). We emphasize that these estimators are for average treatment effects but *not* heterogeneous treatment effects.

### B.2  MODEL-BASED METHODS FOR CONTINUOUS TIME

Orthogonal to our work is a literature stream that focuses on model-based learners for uncertainty quantification or for estimating HTEs over continuous time (Hess et al., 2024b; Hess & Feuerriegel, 2025; Seedat et al., 2022; Vanderschueren et al., 2023). However, our paper focuses on discrete time, and we expect extensions of our meta-learners to continuous time to be an interesting direction for future research.

### B.3  R-LEARNER FOR ESTIMATING HTEs OVER TIME

The R-learner proposed in Lewis & Syrgkanis (2021) is a model-agnostic learner for estimating HTEs over time. In contrast to our meta-learners, it imposes parametric assumptions on the data-generating process. More precisely, the R-learner builds upon the structural nested mean model (SNNM) framework (Robins, 1994), which defines the so-called *blip functions*

$$\gamma_\ell^{\bar{a},\bar{b}}(\bar{h}_t, \bar{x}_{t:t+\ell}, \bar{a}'_{t:t+\ell}) = \mathbb{E}\left[Y_{t:t+\tau}(\bar{a}'_{t:t+\ell}, \bar{a}_{t+\ell+1:t+\tau}) - Y_{t:t+\tau}(\bar{a}'_{t:t+\ell-1}, b_\ell, \bar{a}_{t+\ell+1:t+\tau})\right. \tag{21}$$
$$\left. \mid \bar{H}_t = \bar{h}_t, \bar{X}_{t:t+\ell} = \bar{x}_{t:t+\ell}, \bar{A}_{t:t+\ell} = \bar{a}'_{t:t+\ell}\right]. \tag{22}$$

The main assumption is then that the data-generating process admits a linear blip function

$$\gamma_\ell^{\bar{a},\bar{b}}(\bar{h}_t, \bar{x}_{t:t+\ell}, \bar{a}'_{t:t+\ell}) = \psi_\ell(\bar{h}_t)\phi_\ell^{\bar{a},\bar{b}}(\bar{x}_{t:t+\ell}, \bar{a}'_{t:t+\ell}), \tag{23}$$

where $\phi_\ell^{\bar{a},\bar{b}}$ is known.

**R-learner.** The R-learner proceeds by estimating the nuisance functions

$$q_\ell(\bar{h}_\ell) = \mathbb{E}[Y_{t+\tau} \mid \bar{H}_\ell = \bar{h}_\ell], \tag{24}$$
$$Q_{j,\ell} = \phi_j^{\bar{a},\bar{b}}(\bar{X}_{t:t+j}, \bar{A}_{t:t+j}) - \phi_j^{\bar{a},\bar{b}}(\bar{X}_{t:t+j}, (\bar{a}_{t:t+j-1}, a_{t+j}))\mathbb{1}(j > \ell), \text{ and} \tag{25}$$
$$p_{j,\ell}^{\bar{a}}(\bar{h}_\ell) = \mathbb{E}[Q_{j,\ell} \mid \bar{H}_\ell = \bar{h}_\ell]. \tag{26}$$

Once initial estimators of the nuisance functions are obtained, the R-learner iteratively minimizes the loss

$$\hat{\psi}_\ell = \arg\min_{\psi'_\ell} \hat{\mathbb{E}}_n\left[\left(Y_{t+\tau} - \hat{q}_\ell(\bar{H}_\ell) - \sum_{j=\ell+1}^{t+\tau}\left(Q_{j,\ell} - \hat{p}_{j,\ell}^{\bar{a}}(\bar{H}_\ell)\right)\hat{\psi}_j(\bar{h}_t) - \left(Q_{\ell,\ell} - \hat{p}_{\ell,\ell}^{\bar{a}}(\bar{H}_\ell)\right)\psi'_\ell(\bar{h}_t)\right)^2\right] \tag{27}$$

In the original paper (Lewis & Syrgkanis, 2021), the authors consider estimating HTEs under a linear Markovian assumption, which they show corresponds to setting $\phi_\ell^{\bar{a},\bar{b}}(\bar{x}_{t:t+\ell}, \bar{a}'_{t:t+\ell}) = a'_{t+\ell}$. Then, they estimate CATE via

$$\hat{\tau}_{\mathrm{R}}^{\bar{a},\bar{b}}(\bar{h}_t) = \sum_{\ell=t}^{t+\tau}\hat{\psi}_\ell(\bar{h}_t)(a_\ell - b_\ell). \tag{28}$$

# C PROOFS

## C.1 IDENTIFIABILITY OF CATE AND CAPO

**Lemma 1** (G-formula (Robins, 1999)). *Under Assumption 1, both CAPO and CATE are identified and can be expressed in terms of the observational data distribution.*

*Proof.* Under Assumption 1, we can write the CAPO as

$$\mathbb{E}[Y_{t+\tau}(\bar{a}_{t:t+\tau}) \mid \bar{H}_t = \bar{h}_t]$$

$$\overset{(*)}{=} \mathbb{E}[Y_{t+\tau}(\bar{a}_{t:t+\tau}) \mid \bar{H}_t = \bar{h}_t, A_t = a_t] \tag{29}$$

$$\overset{(**)}{=} \mathbb{E}[\mathbb{E}[Y_{t+\tau}(\bar{a}_{t:t+\tau}) \mid \bar{H}_{t+1} = \bar{h}_{t+1}] \mid \bar{H}_t = \bar{h}_t, A_t = a_t] \tag{30}$$

$$= \mathbb{E}[\mathbb{E}[Y_{t+\tau}(\bar{a}_{t:t+\tau}) \mid \bar{H}_{t+1} = \bar{h}_{t+1}, A_{t:t+1} = a_{t:t+1}] \mid \bar{H}_t = \bar{h}_t, A_t = a_t] \tag{31}$$

$$= \dots$$

$$= \mathbb{E}[\dots \mathbb{E}[\mathbb{E}[Y_{t+\tau}(\bar{a}_{t:t+\tau}) \mid \bar{H}_{t+\tau} = \bar{h}_{t+\tau}, A_{t:t+\tau} = a_{t:t+\tau}] \tag{32}$$

$$\mid \bar{H}_{t+\tau-1} = \bar{h}_{t+\tau-1}, A_{t:t+\tau-1} = a_{t:t+\tau-1}] \mid \dots \mid \bar{H}_t = \bar{h}_t, A_t = a_t] \tag{33}$$

$$\overset{(***)}{=} \mathbb{E}[\dots \mathbb{E}[\mathbb{E}[Y_{t+\tau} \mid \bar{H}_{t+\tau} = \bar{h}_{t+\tau}, A_{t:t+\tau} = a_{t:t+\tau}] \tag{34}$$

$$\mid \bar{H}_{t+\tau-1} = \bar{h}_{t+\tau-1}, A_{t:t+\tau-1} = a_{t:t+\tau-1}] \mid \dots \mid \bar{H}_t = \bar{h}_t, A_t = a_t] \tag{35}$$

$$, \tag{36}$$

where we used sequential ignorability in $(*)$, the tower property in $(**)$, and consistency in $(***)$. The CATE is thus identified as the difference of CAPOs. $\qquad\square$

## C.2 UNBIASEDNESS OF PSEUDO-OUTCOMES

Here, we show the unbiasedness of the pseudo-outcomes that we consider in our main paper when used as a regression target.

• **PI-RA-learner.** Follows directly from Appendix C.1.

• **RA-learner.** It holds that

$$\mathbb{E}\left[Y_{\mathrm{RA}}^{\bar{a},\bar{b}} \mid \bar{H}_t = \bar{h}_t\right] \tag{37}$$

$$= \mathbb{E}\left[\mathbb{1}\{A_t = a_t\} \left(\mu_{t+1}^{\bar{a}}\left(\bar{H}_{t+1}\right) - \mu_t^{\bar{b}}\left(\bar{H}_t\right)\right) + \mathbb{1}\{A_t = b_t\} \left(\mu_t^{\bar{a}}\left(\bar{H}_t\right) - \mu_{t+1}^{\bar{b}}\left(\bar{H}_{t+1}\right)\right) \tag{38}$$

$$+ \mathbb{1}\{A_t \neq a_t\}\mathbb{1}\{A_t \neq b_t\} \left(\mu_t^{\bar{a}}\left(\bar{h}_t\right) - \mu_t^{\bar{b}}\left(\bar{h}_t\right)\right) \bigg| \bar{H}_t = \bar{h}_t\right] \tag{39}$$

$$= \pi_t(a_t \mid \bar{h}_t) \left(\mathbb{E}\left[\mu_{t+1}^{\bar{a}}\left(\bar{H}_{t+1}\right) \mid \bar{H}_t = \bar{h}_t, A_t = a_t\right] - \mu_t^{\bar{b}}\left(\bar{h}_t\right)\right) \tag{40}$$

$$+ \pi_t(b_t \mid \bar{h}_t) \left(\mu_t^{\bar{a}}\left(\bar{h}_t\right) - \mathbb{E}\left[\mu_{t+1}^{\bar{b}}\left(\bar{H}_{t+1}\right) \mid \bar{H}_t = \bar{h}_t, A_t = b_t\right]\right) \tag{41}$$

$$+ (1 - \pi_t(a_t \mid \bar{h}_t)\pi_t(b_t \mid \bar{h}_t)) \left(\mu_t^{\bar{a}}\left(\bar{h}_t\right) - \mu_t^{\bar{b}}\left(\bar{h}_t\right)\right) \tag{42}$$

$$= \pi_t(a_t \mid \bar{h}_t) \left(\mu_t^{\bar{a}}\left(\bar{h}_t\right) - \mu_t^{\bar{b}}\left(\bar{h}_t\right)\right) + \pi_t(b_t \mid \bar{h}_t) \left(\mu_t^{\bar{a}}\left(\bar{h}_t\right) - \mu_t^{\bar{b}}\left(\bar{h}_t\right)\right) \tag{43}$$

$$+ (1 - \pi_t(a_t \mid \bar{h}_t)\pi_t(b_t \mid \bar{h}_t)) \left(\mu_t^{\bar{a}}\left(\bar{h}_t\right) - \mu_t^{\bar{b}}\left(\bar{h}_t\right)\right) \tag{44}$$

$$= \mu_t^{\bar{a}}\left(\bar{h}_t\right) - \mu_t^{\bar{b}}\left(\bar{h}_t\right). \tag{45}$$

- **IPW-learner.** It holds that

$$\mathbb{E}\left[Y_{\mathrm{IPW}}^{\bar{a}} \mid \bar{H}_t = \bar{h}_t\right] \tag{46}$$

$$=\mathbb{E}\left[\prod_{\ell=t}^{t+\tau} \frac{\mathbb{1}\{A_\ell = a_\ell\}}{\pi_\ell(a_\ell \mid \bar{H}_\ell)} Y_{t+\tau} \middle| \bar{H}_t = \bar{h}_t\right] \tag{47}$$

$$=\mathbb{E}\left[\ldots \mathbb{E}\left[\prod_{\ell=t}^{t+\tau} \frac{\mathbb{1}\{A_\ell = a_\ell\}}{\pi_\ell(a_\ell \mid \bar{H}_\ell)} Y_{t+\tau} \middle| \bar{H}_{t+\tau}\right] \ldots \middle| \bar{H}_t = \bar{h}_t\right] \tag{48}$$

$$=\mathbb{E}\left[\ldots \mathbb{E}\left[\prod_{\ell=t}^{t+\tau-1} \frac{\mathbb{1}\{A_\ell = a_\ell\}}{\pi_\ell(a_\ell \mid \bar{H}_\ell)} \mathbb{E}\left[\frac{\mathbb{1}\{A_{t+\tau} = a_{t+\tau}\}}{\pi_{t+\tau}(a_{t+\tau} \mid \bar{H}_{t+\tau})} Y_{t+\tau} \middle| \bar{H}_{t+\tau}\right] \middle| \bar{H}_{t+\tau-1}\right] \ldots \middle| \bar{H}_t = \bar{h}_t\right] \tag{49}$$

$$=\mathbb{E}\left[\ldots \mathbb{E}\left[\prod_{\ell=t}^{t+\tau-1} \frac{\mathbb{1}\{A_\ell = a_\ell\}}{\pi_\ell(a_\ell \mid \bar{H}_\ell)} \mu_{t+\tau}^{\bar{a}}\left(\bar{H}_{t+\tau}\right) \middle| \bar{H}_{t+\tau-1}\right] \ldots \middle| \bar{H}_t = \bar{h}_t\right] \tag{50}$$

$$=\cdots \tag{51}$$

$$=\mathbb{E}\left[\mu_{t+1}^{\bar{a}}\left(\bar{H}_{t+1}\right) \mid \bar{H}_t = \bar{h}_t\right] \tag{52}$$

$$=\mu_t^{\bar{a}}\left(\bar{h}_t\right) \tag{53}$$

and hence

$$\mathbb{E}\left[Y_{\mathrm{IPW}}^{\bar{a},\bar{b}} \mid \bar{H}_t = \bar{h}_t\right] = \mu_t^{\bar{a}}\left(\bar{h}_t\right) - \mu_t^{\bar{b}}\left(\bar{h}_t\right). \tag{54}$$

- **DR-learner.** It holds that

$$\mathbb{E}\left[Y_{\mathrm{DR}}^{\bar{a}} \mid \bar{H}_t = \bar{h}_t\right] \tag{55}$$

$$=\mathbb{E}\left[Y_{\mathrm{IPW}}^{\bar{a}} \mid \bar{H}_t = \bar{h}_t\right] + \mathbb{E}\left[\sum_{k=t}^{t+\tau} \mu_k^{\bar{a}}\left(\bar{H}_k\right)\left(1 - \frac{\mathbb{1}\{A_k = a_k\}}{\pi_k(a_k \mid \bar{H}_k)}\right)\prod_{\ell=t}^{k-1} \frac{\mathbb{1}\{A_\ell = a_\ell\}}{\pi_\ell(a_\ell \mid \bar{H}_\ell)} \middle| \bar{H}_t = \bar{h}_t\right] \tag{56}$$

$$=\mu_t^{\bar{a}}\left(\bar{h}_t\right) + \mathbb{E}\left[\ldots \mathbb{E}\left[\sum_{k=t}^{t+\tau} \mu_k^{\bar{a}}\left(\bar{H}_k\right)\left(1 - \frac{\mathbb{1}\{A_k = a_k\}}{\pi_k(a_k \mid \bar{H}_k)}\right)\prod_{\ell=t}^{k-1} \frac{\mathbb{1}\{A_\ell = a_\ell\}}{\pi_\ell(a_\ell \mid \bar{H}_\ell)} \middle| \bar{H}_{t+\tau}\right] \ldots \middle| \bar{H}_t = \bar{h}_t\right] \tag{57}$$

$$=\mu_t^{\bar{a}}\left(\bar{h}_t\right) + \mathbb{E}\left[\ldots \mathbb{E}\left[\sum_{k=t}^{t+\tau-1} \mu_k^{\bar{a}}\left(\bar{H}_k\right)\left(1 - \frac{\mathbb{1}\{A_k = a_k\}}{\pi_k(a_k \mid \bar{H}_k)}\right)\prod_{\ell=t}^{k-1} \frac{\mathbb{1}\{A_\ell = a_\ell\}}{\pi_\ell(a_\ell \mid \bar{H}_\ell)} \middle| \bar{H}_{t+\tau-1}\right] \ldots \middle| \bar{H}_t = \bar{h}_t\right] \tag{58}$$

$$=\mu_t^{\bar{a}}\left(\bar{h}_t\right) + \mathbb{E}\left[\mu_t^{\bar{a}}\left(\bar{H}_t\right)\left(1 - \frac{\mathbb{1}\{A_t = a_t\}}{\pi_t(a_t \mid \bar{H}_t)}\right) \middle| \bar{H}_t = \bar{h}_t\right] \tag{59}$$

$$=\mu_t^{\bar{a}}\left(\bar{h}_t\right) \tag{60}$$

and hence

$$\mathbb{E}\left[Y_{\mathrm{DR}}^{\bar{a},\bar{b}} \mid \bar{H}_t = \bar{h}_t\right] = \mu_t^{\bar{a}}\left(\bar{h}_t\right) - \mu_t^{\bar{b}}\left(\bar{h}_t\right). \tag{61}$$

### C.3 PROOF OF THEOREM 2

*Proof.* Without loss of generality, we show the result for $Y_{\mathrm{DR}}^{\bar{a},\bar{b}}$. The result for $Y_{\mathrm{DR}}^{\bar{a}}$ follows analogously. By iteratively applying the law of total variance, we obtain

$$\mathrm{Var}(Y_{\mathrm{DR}}^{\bar{a},\bar{b}} \mid \bar{H}_t = \bar{h}_t) \tag{62}$$

$$= \mathbb{E}\left[\mathrm{Var}(Y_{\mathrm{DR}}^{\bar{a},\bar{b}} \mid \bar{H}_{t+\tau}, A_{t+\tau}) \mid \bar{H}_t = \bar{h}_t\right] \tag{63}$$

$$\quad + \mathrm{Var}\left(\mathbb{E}[Y_{\mathrm{DR}}^{\bar{a},\bar{b}} \mid \bar{H}_{t+\tau}, A_{t+\tau}] \mid \bar{H}_t = \bar{h}_t\right) \tag{64}$$

$$= \mathbb{E}\left[\mathrm{Var}(Y_{\mathrm{DR}}^{\bar{a},\bar{b}} \mid \bar{H}_{t+\tau}, A_{t+\tau}) \mid \bar{H}_t = \bar{h}_t\right] \tag{65}$$

$$+ \mathbb{E}\left[ \mathrm{Var}\left( \mathbb{E}[Y_{\mathrm{DR}}^{\bar{a},\bar{b}} \mid \bar{H}_{t+\tau}, A_{t+\tau}) \mid \bar{H}_{t+\tau-1}, A_{t+\tau-1} \right) \mid \bar{H}_t = \bar{h}_t \right] \tag{66}$$

$$+ \mathrm{Var}\left( \mathbb{E}\left[ \mathbb{E}[Y_{\mathrm{DR}}^{\bar{a},\bar{b}} \mid \bar{H}_{t+\tau}, A_{t+\tau}] \mid \bar{H}_{t+\tau-1}, A_{t+\tau-1} \right] \mid \bar{H}_t = \bar{h}_t \right) \tag{67}$$

$$= \ldots \tag{68}$$

$$= \sum_{\ell=t}^{t+\tau} \mathbb{E}\left[ \mathrm{Var}\left( \mathbb{E}\left[ \ldots \mathbb{E}[Y_{\mathrm{DR}}^{\bar{a},\bar{b}} \mid \bar{H}_{t+\tau}, A_{t+\tau}] \ldots \mid \bar{H}_{k+1}, A_{k+1} \right] \mid \bar{H}_k, A_k \right) \mid \bar{H}_t = \bar{h}_t \right] \tag{69}$$

$$+ \mathrm{Var}\left( \mathbb{E}\left[ \ldots \mathbb{E}[Y_{\mathrm{DR}}^{\bar{a},\bar{b}} \mid \bar{H}_{t+\tau}, A_{t+\tau}] \ldots \mid \bar{H}_t, A_t \right] \mid \bar{H}_t = \bar{h}_t \right). \tag{70}$$

For the variance term, we obtain

$$\mathbb{E}\left[ \ldots \mathbb{E}[Y_{\mathrm{DR}}^{\bar{a},\bar{b}} \mid \bar{H}_{t+\tau}, A_{t+\tau}] \ldots \mid \bar{H}_t, A_t \right] \tag{71}$$

$$= \mathbb{E}\left[ \ldots \mathbb{E}\left[ Y_{\mathrm{IPW}}^{\bar{a},\bar{b}} + \sum_{k=t}^{t+\tau} \mu_k^{\bar{a}}\left( \bar{H}_k \right)\left( 1 - \frac{\mathbb{1}\{A_k = a_k\}}{\pi_k(a_k \mid \bar{H}_k)} \right) \prod_{\ell=t}^{k-1} \frac{\mathbb{1}\{A_\ell = a_\ell\}}{\pi_\ell(a_\ell \mid \bar{H}_\ell)} \right.\right. \tag{72}$$

$$\left.\left. - \sum_{k=t}^{t+\tau} \mu_k^{\bar{b}}\left( \bar{H}_k \right)\left( 1 - \frac{\mathbb{1}\{A_k = b_k\}}{\pi_k(b_k \mid \bar{H}_k)} \right) \prod_{\ell=t}^{k-1} \frac{\mathbb{1}\{A_\ell = b_\ell\}}{\pi_\ell(b_\ell \mid \bar{H}_\ell)} \mid \bar{H}_{t+\tau}, A_{t+\tau} \right] \ldots \mid \bar{H}_t, A_t \right] \tag{73}$$

$$= \mathbb{E}\left[ \ldots \mathbb{E}\left[ \underbrace{\left( \prod_{\ell=t}^{t+\tau} \frac{\mathbb{1}\{A_\ell = a_\ell\}}{\pi_\ell(a_\ell \mid \bar{H}_\ell)} \right)\left( \mu_{t+\tau}^{\bar{A}}\left( \bar{H}_{t+\tau} \right) - \mu_{t+\tau}^{\bar{a}}\left( \bar{H}_{t+\tau} \right) \right)}_{=0} \right.\right. \tag{74}$$

$$\underbrace{- \left( \prod_{\ell=t}^{t+\tau} \frac{\mathbb{1}\{A_\ell = b_\ell\}}{\pi_\ell(b_\ell \mid \bar{H}_\ell)} \right)\left( \mu_{t+\tau}^{\bar{A}}\left( \bar{H}_{t+\tau} \right) - \mu_{t+\tau}^{\bar{b}}\left( \bar{H}_{t+\tau} \right) \right)}_{=0} \tag{75}$$

$$+ \left( \prod_{\ell=t}^{t+\tau-1} \frac{\mathbb{1}\{A_\ell = a_\ell\}}{\pi_\ell(a_\ell \mid \bar{H}_\ell)} \right) \mu_{t+\tau}^{\bar{a}}\left( \bar{H}_{t+\tau} \right) - \left( \prod_{\ell=t}^{t+\tau-1} \frac{\mathbb{1}\{A_\ell = b_\ell\}}{\pi_\ell(b_\ell \mid \bar{H}_\ell)} \right) \mu_{t+\tau}^{\bar{b}}\left( \bar{H}_{t+\tau} \right) \tag{76}$$

$$+ \sum_{k=t}^{t+\tau-1} \mu_k^{\bar{a}}\left( \bar{H}_k \right)\left( 1 - \frac{\mathbb{1}\{A_k = a_k\}}{\pi_k(a_k \mid \bar{H}_k)} \right) \prod_{\ell=t}^{k-1} \frac{\mathbb{1}\{A_\ell = a_\ell\}}{\pi_\ell(a_\ell \mid \bar{H}_\ell)} \tag{77}$$

$$\left.\left. - \sum_{k=t}^{t+\tau-1} \mu_k^{\bar{b}}\left( \bar{H}_k \right)\left( 1 - \frac{\mathbb{1}\{A_k = b_k\}}{\pi_k(b_k \mid \bar{H}_k)} \right) \prod_{\ell=t}^{k-1} \frac{\mathbb{1}\{A_\ell = b_\ell\}}{\pi_\ell(b_\ell \mid \bar{H}_\ell)} \mid \bar{H}_{t+\tau-1}, A_{t+\tau-1} \right] \ldots \mid \bar{H}_t, A_t \right] \tag{78}$$

$$= \mathbb{E}\left[ \ldots \mathbb{E}\left[ \underbrace{\left( \prod_{\ell=t}^{t+\tau-1} \frac{\mathbb{1}\{A_\ell = a_\ell\}}{\pi_\ell(a_\ell \mid \bar{H}_\ell)} \right)\left( \mu_{t+\tau-1}^{\bar{A}}\left( \bar{H}_{t+\tau-1} \right) - \mu_{t+\tau-1}^{\bar{a}}\left( \bar{H}_{t+\tau-1} \right) \right)}_{=0} \right.\right. \tag{79}$$

$$\underbrace{- \left( \prod_{\ell=t}^{t+\tau-1} \frac{\mathbb{1}\{A_\ell = b_\ell\}}{\pi_\ell(b_\ell \mid \bar{H}_\ell)} \right)\left( \mu_{t+\tau-1}^{\bar{A}}\left( \bar{H}_{t+\tau-1} \right) - \mu_{t+\tau-1}^{\bar{b}}\left( \bar{H}_{t+\tau-1} \right) \right)}_{=0} \tag{80}$$

$$+ \left( \prod_{\ell=t}^{t+\tau-2} \frac{\mathbb{1}\{A_\ell = a_\ell\}}{\pi_\ell(a_\ell \mid \bar{H}_\ell)} \right) \mu_{t+\tau-1}^{\bar{a}}\left( \bar{H}_{t+\tau-1} \right) - \left( \prod_{\ell=t}^{t+\tau-2} \frac{\mathbb{1}\{A_\ell = b_\ell\}}{\pi_\ell(b_\ell \mid \bar{H}_\ell)} \right) \mu_{t+\tau-1}^{\bar{b}}\left( \bar{H}_{t+\tau-1} \right) \tag{81}$$

$$+ \sum_{k=t}^{t+\tau-2} \mu_k^{\bar{a}}\left( \bar{H}_k \right)\left( 1 - \frac{\mathbb{1}\{A_k = a_k\}}{\pi_k(a_k \mid \bar{H}_k)} \right) \prod_{\ell=t}^{k-1} \frac{\mathbb{1}\{A_\ell = a_\ell\}}{\pi_\ell(a_\ell \mid \bar{H}_\ell)} \tag{82}$$

$$-\sum_{k=t}^{t+\tau-2} \mu_k^{\bar{b}}\left(\bar{H}_k\right)\left(1-\frac{\mathbb{1}\{A_k=b_k\}}{\pi_k(b_k \mid \bar{H}_k)}\right)\prod_{\ell=t}^{k-1}\frac{\mathbb{1}\{A_\ell=b_\ell\}}{\pi_\ell(b_\ell \mid \bar{H}_\ell)} \mid \bar{H}_{t+\tau-2}, A_{t+\tau-2}\Bigg]\dots \Bigg| \bar{H}_t, A_t\Bigg] \tag{83}$$

$$= \dots \tag{84}$$

$$= \underbrace{\frac{\mathbb{1}\{A_t=a_t\}}{\pi_t(a_t \mid \bar{H}_t)}\left(\mu_t^{\bar{A}}\left(\bar{H}_t\right)-\mu_t^{\bar{a}}\left(\bar{H}_t\right)\right)-\frac{\mathbb{1}\{A_\ell=b_\ell\}}{\pi_\ell(b_\ell \mid \bar{H}_\ell)}\left(\mu_{t+1}^{\bar{A}}\left(\bar{H}_t\right)-\mu_t^{\bar{b}}\left(\bar{H}_t\right)\right)}_{=0} \tag{85}$$

$$+\quad \mu_t^{\bar{a}}\left(\bar{H}_t\right)-\mu_t^{\bar{b}}\left(\bar{H}_t\right) \tag{86}$$

$$= \mu_t^{\bar{a}}\left(\bar{H}_t\right)-\mu_t^{\bar{b}}\left(\bar{H}_t\right), \tag{87}$$

which implies

$$\mathrm{Var}\left(\mathbb{E}\left[\dots\mathbb{E}[Y_{\mathrm{DR}}^{\bar{a},\bar{b}} \mid \bar{H}_{t+\tau}, A_{t+\tau}]\dots \mid \bar{H}_t, A_t\right] \Bigg| \bar{H}_t=\bar{h}_t\right) \tag{88}$$

$$= \mathrm{Var}\left(\mu_t^{\bar{a}}\left(\bar{H}_t\right)-\mu_t^{\bar{b}}\left(\bar{H}_t\right) \Bigg| \bar{H}_t=\bar{h}_t\right)=0. \tag{89}$$

Furthermore,

$$\mathrm{Var}\left(\mathbb{E}\left[\dots\mathbb{E}[Y_{\mathrm{DR}}^{\bar{a},\bar{b}} \mid \bar{H}_{t+\tau}, A_{t+\tau}]\dots \mid \bar{H}_{k+1}, A_{k+1}\right] \Bigg| \bar{H}_k, A_k\right) \tag{90}$$

$$\stackrel{(*)}{=} \mathrm{Var}\left(\sum_{j=t}^{k+1}\left(\prod_{\ell=t}^{j-1}\frac{\mathbb{1}\{A_\ell=a_\ell\}}{\pi_\ell(a_\ell \mid \bar{H}_\ell)}\right)\left(\mu_j^{\bar{a}}\left(\bar{H}_j\right)-\mu_{j-1}^{\bar{a}}\left(\bar{H}_{j-1}\right)\right)\right. \tag{91}$$

$$\left.-\sum_{j=t}^{k+1}\left(\prod_{\ell=t}^{j-1}\frac{\mathbb{1}\{A_\ell=b_\ell\}}{\pi_\ell(b_\ell \mid \bar{H}_\ell)}\right)\left(\mu_j^{\bar{b}}\left(\bar{H}_j\right)-\mu_{j-1}^{\bar{b}}\left(\bar{H}_{j-1}\right)\right) \Bigg| \bar{H}_k, A_k\right) \tag{92}$$

$$= \mathrm{Var}\left(\left(\prod_{\ell=t}^{k}\frac{\mathbb{1}\{A_\ell=a_\ell\}}{\pi_\ell(a_\ell \mid \bar{H}_\ell)}\right)\left(\mu_{k+1}^{\bar{a}}\left(\bar{H}_{k+1}\right)-\mu_k^{\bar{a}}\left(\bar{H}_k\right)\right)\right. \tag{93}$$

$$\left.-\left(\prod_{\ell=t}^{k}\frac{\mathbb{1}\{A_\ell=b_\ell\}}{\pi_\ell(b_\ell \mid \bar{H}_\ell)}\right)\left(\mu_{k+1}^{\bar{b}}\left(\bar{H}_{k+1}\right)-\mu_k^{\bar{b}}\left(\bar{H}_k\right)\right) \Bigg| \bar{H}_k, A_k\right) \tag{94}$$

$$= \left(\prod_{\ell=t}^{k}\frac{\mathbb{1}\{A_\ell=a_\ell\}}{\pi_\ell^2(a_\ell \mid \bar{H}_\ell)}\right)\mathrm{Var}\left(\mu_{k+1}^{\bar{a}}\left(\bar{H}_{k+1}\right) \mid \bar{H}_k, A_k\right) \tag{95}$$

$$+ \left(\prod_{\ell=t}^{k}\frac{\mathbb{1}\{A_\ell=b_\ell\}}{\pi_\ell^2(b_\ell \mid \bar{H}_\ell)}\right)\mathrm{Var}\left(\mu_{k+1}^{\bar{b}}\left(\bar{H}_{k+1}\right) \mid \bar{H}_k, A_k\right) \tag{96}$$

$$= \sigma^2\left(\prod_{\ell=t}^{k}\frac{\mathbb{1}\{A_\ell=a_\ell\}}{\pi_\ell^2(a_\ell \mid \bar{H}_\ell)}+\prod_{\ell=t}^{k}\frac{\mathbb{1}\{A_\ell=b_\ell\}}{\pi_\ell^2(b_\ell \mid \bar{H}_\ell)}\right), \tag{97}$$

where $(*)$ follows from the same arguments from Eq. 71 onwards.

Hence,

$$\mathrm{Var}(Y_{\mathrm{DR}}^{\bar{a},\bar{b}} \mid \bar{H}_t=\bar{h}_t) \tag{98}$$

$$= \mathbb{E}\left[\sum_{\ell=t}^{t+\tau}\mathrm{Var}\left(\mathbb{E}\left[\dots\mathbb{E}[Y_{\mathrm{DR}}^{\bar{a},\bar{b}} \mid \bar{H}_{t+\tau}, A_{t+\tau}]\dots \mid \bar{H}_{k+1}, A_{k+1}\right] \mid \bar{H}_k, A_k\right) \Bigg| \bar{H}_t=\bar{h}_t\right] \tag{99}$$

$$= \sigma^2\mathbb{E}\left[\sum_{\ell=t}^{t+\tau}\left(\prod_{k=t}^{\ell}\frac{\mathbb{1}\{A_k=a_k\}}{\pi_k^2(a_k \mid \bar{H}_k)}+\prod_{k=t}^{\ell}\frac{\mathbb{1}\{A_k=b_k\}}{\pi_k^2(b_k \mid \bar{H}_k)}\right) \Bigg| \bar{H}_t=\bar{h}_t\right] \tag{100}$$

$$= \sigma^2 \mathbb{E}\left[V_{\text{DR}}^{\bar{a},\bar{b}} \mid \bar{H}_t = \bar{h}_t\right]. \tag{101}$$

$\square$

## C.4 PROOF OF THEOREM 1

Before proving Theorem 1, we impose additional assumptions from Kennedy (2023).

**Assumption 5** (from (Kennedy, 2023)). Let $\hat{\mathbb{E}}_n$ be a regression estimator for either the response functions or the pseudo-outcome regressions. Furthermore, we denote $Y_{\text{tar}}$ as the target variable and $Z$ as the conditioning variables of the corresponding regression. We assume $\hat{Y}_{\text{tar}} \xrightarrow{p} Y_{\text{tar}}$ (consistency of the target estimator) and that any regression estimator $\hat{\mathbb{E}}_n$ satisfies

$$\frac{\hat{\mathbb{E}}_n[\hat{Y}_{\text{tar}} \mid Z = z] - \hat{\mathbb{E}}_n[Y_{\text{tar}} \mid Z = z] - \hat{\mathbb{E}}_n[\hat{Y}_{\text{tar}} - Y_{\text{tar}} \mid Z = z]}{\sqrt{\mathbb{E}\left[\left(\hat{\mathbb{E}}_n[Y_{\text{tar}} \mid Z = z] - \mathbb{E}[Y_{\text{tar}} \mid Z = z]\right)^2\right]}} \xrightarrow{p} 0 \tag{102}$$

and

$$\mathbb{E}\left[\hat{\mathbb{E}}_n[\hat{b}(Z) \mid Z = z]^2\right] \lesssim \mathbb{E}\left[\hat{b}(z)^2\right], \tag{103}$$

where $\hat{b}(z) = \mathbb{E}[\hat{Y}_{\text{tar}} - Y_{\text{tar}} \mid Z = z]$.

Kennedy (2023) showed that these assumptions hold, e.g., for linear smoothers. Assumption 5 implies the following lemma.

**Lemma 2.** *Under Assumption 5, for any regression estimator $\hat{\mathbb{E}}_n$, it holds that*

$$\mathbb{E}[(\hat{\mathbb{E}}_n[\hat{Y}_{\text{tar}} \mid Z = z] - \mathbb{E}[Y_{\text{tar}} \mid Z = z])^2] \lesssim r_{\text{PO}}(n) + \mathbb{E}\left[\hat{b}(z)^2\right]. \tag{104}$$

We now proceed with the proof of Theorem 1.

*Proof.* We prove the rates for each learner separately.

**PI-HA-learner.** Let us define

$$\widetilde{\tau}_{\text{PI-HA}}^{\bar{a},\bar{b}}(\bar{h}_t) = \mathbb{E}\left[Y_{t+\tau} \mid \bar{H}_t = \bar{h}_t, \bar{A}_{t:t+\tau} = \bar{a}_{t:t+\tau}\right] - \mathbb{E}\left[Y_{t+\tau} \mid \bar{H}_t = \bar{h}_t, \bar{A}_{t:t+\tau} = \bar{b}_{t:t+\tau}\right], \tag{105}$$

which is the target estimand of the PI-HA-learner in population. We then can write

$$\mathbb{E}[(\hat{\tau}_{\text{PI-HA}}^{\bar{a},\bar{b}}(\bar{h}_t) - \tau_{\bar{a},\bar{b}}(\bar{h}_t))^2] \tag{106}$$

$$= \mathbb{E}\left[(\hat{\tau}_{\text{PI-HA}}^{\bar{a},\bar{b}}(\bar{h}_t) - \widetilde{\tau}_{\text{PI-HA}}^{\bar{a},\bar{b}}(\bar{h}_t) + \widetilde{\tau}_{\text{PI-HA}}^{\bar{a},\bar{b}}(\bar{h}_t) - \tau_{\bar{a},\bar{b}}(\bar{h}_t))^2\right] \tag{107}$$

$$\leq \mathbb{E}\left[(\hat{\tau}_{\text{PI-HA}}^{\bar{a},\bar{b}}(\bar{h}_t) - \widetilde{\tau}_{\text{PI-HA}}^{\bar{a},\bar{b}}(\bar{h}_t))^2\right] + \text{bias}_{\bar{a},\bar{b}}^2 \tag{108}$$

$$\leq \mathbb{E}\left[(\mathbb{E}\left[Y_{t+\tau} \mid \bar{H}_t = \bar{h}_t, \bar{A}_{t:t+\tau} = \bar{a}_{t:t+\tau}\right] - \hat{\mathbb{E}}_n\left[Y_{t+\tau} \mid \bar{H}_t = \bar{h}_t, \bar{A}_{t:t+\tau} = \bar{a}_{t:t+\tau}\right])^2\right] \tag{109}$$

$$+ \mathbb{E}\left[(\mathbb{E}\left[Y_{t+\tau} \mid \bar{H}_t = \bar{h}_t, \bar{A}_{t:t+\tau} = \bar{b}_{t:t+\tau}\right] - \hat{\mathbb{E}}_n\left[Y_{t+\tau} \mid \bar{H}_t = \bar{h}_t, \bar{A}_{t:t+\tau} = \bar{b}_{t:t+\tau}\right])^2\right] + \text{bias}_{\bar{a},\bar{b}}^2 \tag{110}$$

$$\lesssim r_{\text{PI-HA}}^{\bar{a}}(n) + r_{\text{PI-HA}}^{\bar{b}}(n) + \text{bias}_{\bar{a},\bar{b}}^2, \tag{111}$$

which proves the rate for the PI-HA-learner.

**PI-HA-learner.** By applying Lemma 2 iteratively, we obtain

$$\mathbb{E}\left[\left(\hat{\mu}_t^{\bar{a}}(\bar{h}_t) - \mu_t^{\bar{a}}(\bar{h}_t)\right)^2\right] \lesssim r_{\mu_t}^{\bar{a}}(n) + \mathbb{E}\left[\mathbb{E}\left[\hat{\mu}_{t+1}^{\bar{a}}(\bar{H}_{t+1}) - \mu_{t+1}^{\bar{a}}(\bar{H}_{t+1}) \mid \bar{H}_t = \bar{h}_t\right]^2\right] \tag{112}$$

$$\leq r_{\mu_t}^{\bar{a}}(n) + \mathbb{E}\left[\mathbb{E}\left[\left(\hat{\mu}_{t+1}^{\bar{a}}(\bar{H}_{t+1}) - \mu_{t+1}^{\bar{a}}(\bar{H}_{t+1})\right)^2\right] \mid \bar{H}_t = \bar{h}_t\right] \tag{113}$$

$$\lesssim \cdots \tag{114}$$

$$\lesssim \sum_{\ell=t}^{t+\tau} r_{\mu_\ell}^{\bar{a}}(n). \tag{115}$$

The result for CATE follows by noting that

$$\mathbb{E}[(\hat{\tau}_{\text{PI-RA}}^{\bar{a},\bar{b}}(\bar{h}_t) - \tau_{\bar{a},\bar{b}}(\bar{h}_t))^2] \leq \mathbb{E}\left[\left(\hat{\mu}_t^{\bar{a}}(\bar{h}_t) - \mu_t^{\bar{a}}(\bar{h}_t)\right)^2\right] + \mathbb{E}\left[\left(\hat{\mu}_t^{\bar{b}}(\bar{h}_t) - \mu_t^{\bar{b}}(\bar{h}_t)\right)^2\right]. \tag{116}$$

**RA-learner.** We start by deriving a formula for the bias term via

$$\hat{b}_{\text{RA}}(\bar{h}_t) = \pi_t(a_t \mid \bar{h}_t)\left(\mathbb{E}\left[\hat{\mu}_{t+1}^{\bar{a}}\left(\bar{H}_{t+1}\right) \mid \bar{H}_t = \bar{h}_t\right] - \mu_t^{\bar{a}}\left(\bar{h}_t\right) + \mu_t^{\bar{b}}\left(\bar{h}_t\right) - \hat{\mu}_t^{\bar{b}}\left(\bar{h}_t\right)\right) \tag{117}$$

$$+ \pi_t(b_t \mid \bar{h}_t)\left(\hat{\mu}_t^{\bar{a}}\left(\bar{h}_t\right) - \mu_t^{\bar{a}}\left(\bar{h}_t\right) + \mu_t^{\bar{b}}\left(\bar{h}_t\right) - \mathbb{E}\left[\hat{\mu}_{t+1}^{\bar{b}}\left(\bar{H}_{t+1}\right) \mid \bar{H}_t = \bar{h}_t\right]\right) \tag{118}$$

$$+ \left(1 - \pi_t(a_t \mid \bar{h}_t) - \pi_t(b_t \mid \bar{h}_t)\right)\left(\hat{\mu}_t^{\bar{a}}\left(\bar{h}_t\right) - \mu_t^{\bar{a}}\left(\bar{h}_t\right) + \mu_t^{\bar{b}}\left(\bar{h}_t\right) - \hat{\mu}_t^{\bar{b}}\left(\bar{h}_t\right)\right). \tag{119}$$

Hence, we can apply Lemma 2 and obtain

$$\mathbb{E}[(\hat{\tau}_{\text{RA}}^{\bar{a},\bar{b}}(\bar{h}_t) - \tau_{\bar{a},\bar{b}}(\bar{h}_t))^2] \lesssim r_{\text{PO}}^{\bar{a},\bar{b}}(n) + \mathbb{E}\left[\hat{b}_{\text{RA}}(\bar{h}_t)^2\right] \tag{120}$$

$$\lesssim r_{\text{PO}}^{\bar{a},\bar{b}}(n) + \pi_t^2(a_t \mid \bar{h}_t)\left(\sum_{\ell=t+1}^{t+\tau} r_{\mu_\ell}^{\bar{a}}(n) + \sum_{\ell=t}^{t+\tau} r_{\mu_\ell}^{\bar{b}}(n)\right) \tag{121}$$

$$+ \pi_t^2(b_t \mid \bar{h}_t)\left(\sum_{\ell=t}^{t+\tau} r_{\mu_\ell}^{\bar{a}}(n) + \sum_{\ell=t+1}^{t+\tau} r_{\mu_\ell}^{\bar{b}}(n)\right) \tag{122}$$

$$+ \left(1 - \pi_t(a_t \mid \bar{h}_t) - \pi_t(b_t \mid \bar{h}_t)\right)^2\left(\sum_{\ell=t}^{t+\tau} r_{\mu_\ell}^{\bar{a}}(n) + \sum_{\ell=t+1}^{t+\tau} r_{\mu_\ell}^{\bar{b}}(n)\right) \tag{123}$$

$$\lesssim r_{\text{PO}}^{\bar{a},\bar{b}}(n) + \sum_{\ell=t}^{t+\tau} r_{\mu_\ell}^{\bar{a}}(n) + \sum_{\ell=t+1}^{t+\tau} r_{\mu_\ell}^{\bar{b}}(n). \tag{124}$$

**IPW learner.** The bias term for the IPW-learner can be written as

$$\hat{b}_{\text{IPW}}^{\bar{a}}(\bar{h}_t) \tag{125}$$

$$= \mathbb{E}\left[\left(\prod_{\ell=t}^{t+\tau} \frac{\mathbb{1}\{A_\ell = a_\ell\}}{\hat{\pi}_\ell(a_\ell \mid \bar{H}_\ell)} - \prod_{\ell=t}^{t+\tau} \frac{\mathbb{1}\{A_\ell = a_\ell\}}{\pi_\ell(a_\ell \mid \bar{H}_\ell)}\right) Y_{t+\tau} \mid \bar{H}_t = \bar{h}_t\right] \tag{126}$$

$$= \mathbb{E}\left[\dots\mathbb{E}\left[\left(\prod_{\ell=t}^{t+\tau} \frac{\mathbb{1}\{A_\ell = a_\ell\}}{\hat{\pi}_\ell(a_\ell \mid \bar{H}_\ell)} - \prod_{\ell=t}^{t+\tau} \frac{\mathbb{1}\{A_\ell = a_\ell\}}{\pi_\ell(a_\ell \mid \bar{H}_\ell)}\right) Y_{t+\tau} \mid \bar{H}_{t+\tau}\right]\dots \mid \bar{H}_t = \bar{h}_t\right] \tag{127}$$

$$= \mathbb{E}\left[\dots\mathbb{E}\left[\left(\prod_{\ell=t}^{t+\tau-1} \frac{\mathbb{1}\{A_\ell = a_\ell\}}{\hat{\pi}_\ell(a_\ell \mid \bar{H}_\ell)} \frac{\pi_{t+\tau}(a_{t+\tau} \mid \bar{H}_{t+\tau})}{\hat{\pi}_{t+\tau}(a_{t+\tau} \mid \bar{H}_{t+\tau})}\right.\right.\right. \tag{128}$$

$$\left.\left.\left. - \prod_{\ell=t}^{t+\tau-1} \frac{\mathbb{1}\{A_\ell = a_\ell\}}{\pi_\ell(a_\ell \mid \bar{H}_\ell)} \frac{\pi_{t+\tau}(a_{t+\tau} \mid \bar{H}_{t+\tau})}{\pi_{t+\tau}(a_{t+\tau} \mid \bar{H}_{t+\tau})}\right) \mu_{t+\tau}^{\bar{a}}\left(\bar{H}_{t+\tau}\right) \mid \bar{H}_{t+\tau-1}\right]\dots \mid \bar{H}_t = \bar{h}_t\right] \tag{129}$$

$$= \mathbb{E}\left[\dots\mathbb{E}\left[\left(\prod_{\ell=t}^{t+\tau-1} \frac{\mathbb{1}\{A_\ell = a_\ell\}}{\hat{\pi}_\ell(a_\ell \mid \bar{H}_\ell)} \frac{\pi_{t+\tau}(a_{t+\tau} \mid \bar{H}_{t+\tau})}{\hat{\pi}_{t+\tau}(a_{t+\tau} \mid \bar{H}_{t+\tau})}\right.\right.\right. \tag{130}$$

$$\left. - \prod_{\ell=t}^{t+\tau-1} \frac{\mathbb{1}\{A_\ell = a_\ell\}}{\hat{\pi}_\ell(a_\ell \mid \bar{H}_\ell)} \frac{\pi_{t+\tau}(a_{t+\tau} \mid \bar{H}_{t+\tau})}{\pi_{t+\tau}(a_{t+\tau} \mid \bar{H}_{t+\tau})} + \prod_{\ell=t}^{t+\tau-1} \frac{\mathbb{1}\{A_\ell = a_\ell\}}{\hat{\pi}_\ell(a_\ell \mid \bar{H}_\ell)} \frac{\pi_{t+\tau}(a_{t+\tau} \mid \bar{H}_{t+\tau})}{\pi_{t+\tau}(a_{t+\tau} \mid \bar{H}_{t+\tau})}\right. \tag{131}$$

$$\left.\left.\left. - \prod_{\ell=t}^{t+\tau-1} \frac{\mathbb{1}\{A_\ell = a_\ell\}}{\pi_\ell(a_\ell \mid \bar{H}_\ell)} \frac{\pi_{t+\tau}(a_{t+\tau} \mid \bar{H}_{t+\tau})}{\pi_{t+\tau}(a_{t+\tau} \mid \bar{H}_{t+\tau})}\right) \mu_{t+\tau}^{\bar{a}}\left(\bar{H}_{t+\tau}\right) \mid \bar{H}_{t+\tau-1}\right]\dots \mid \bar{H}_t = \bar{h}_t\right] \tag{132}$$

$$= \mathbb{E}\left[\prod_{\ell=t}^{t+\tau-1} \frac{\mathbb{1}\{A_\ell = a_\ell\}}{\hat{\pi}_\ell(a_\ell \mid \bar{H}_\ell)} \frac{\pi_{t+\tau}(a_{t+\tau} \mid \bar{H}_{t+\tau}) - \hat{\pi}_{t+\tau}(a_{t+\tau} \mid \bar{H}_{t+\tau})}{\hat{\pi}_{t+\tau}(a_{t+\tau} \mid \bar{H}_{t+\tau})} \mu_{t+\tau}^{\bar{a}}\left(\bar{H}_{t+\tau}\right) \mid \bar{H}_t = \bar{h}_t\right] \tag{133}$$

$$+ \mathbb{E}\left[\ldots \mathbb{E}\left[\left(\prod_{\ell=t}^{t+\tau-1} \frac{\mathbb{1}\{A_\ell = a_\ell\}}{\hat{\pi}_\ell(a_\ell \mid \bar{H}_\ell)} - \prod_{\ell=t}^{t+\tau-1} \frac{\mathbb{1}\{A_\ell = a_\ell\}}{\pi_\ell(a_\ell \mid \bar{H}_\ell)}\right) \mu_{t+\tau}^{\bar{a}}\left(\bar{H}_{t+\tau}\right) \mid \bar{H}_{t+\tau-1}\right]\ldots \mid \bar{H}_t = \bar{h}_t\right] \tag{134}$$

$$= \ldots \tag{135}$$

$$= \sum_{k=t}^{t+\tau} \mathbb{E}\left[\prod_{\ell=t}^{k-1} \frac{\mathbb{1}\{A_\ell = a_\ell\}}{\hat{\pi}_\ell(a_\ell \mid \bar{H}_\ell)} \frac{\pi_k(a_k \mid \bar{H}_k) - \hat{\pi}_k(a_k \mid \bar{H}_k)}{\hat{\pi}_k(a_k \mid \bar{H}_k)} \mu_k^{\bar{a}}\left(\bar{H}_k\right) \mid \bar{H}_t = \bar{h}_t\right]. \tag{136}$$

Hence, Lemma 2 yields

$$\mathbb{E}\left[\hat{b}_{\mathrm{IPW}}^{\bar{a}}(\bar{h}_t)^2\right] \overset{(*)}{\leq} C\,\mathbb{E}\left[\sum_{k=t}^{t+\tau} \mathbb{E}\left[\left(\pi_k(a_k \mid \bar{H}_k) - \hat{\pi}_k(a_k \mid \bar{H}_k)\right)^2 \mid \bar{H}_t = \bar{h}_t\right]\right] \tag{137}$$

$$\overset{(**)}{=} C \sum_{k=t}^{t+\tau} \mathbb{E}\left[\mathbb{E}\left[\left(\pi_k(a_k \mid \bar{H}_k) - \hat{\pi}_k(a_k \mid \bar{H}_k)\right)^2\right] \mid \bar{H}_t = \bar{h}_t\right] \tag{138}$$

$$\lesssim \sum_{\ell=t}^{t+\tau} r_{\pi_\ell}^{\bar{a}}(n), \tag{139}$$

where $(*)$ follows from the boundedness assumptions (2) and $(**)$ from applying Fubini's theorem. We obtain the rate for CATE via

$$\mathbb{E}[(\hat{\tau}_{\mathrm{IPW}}^{\bar{a},\bar{b}}(\bar{h}_t) - \tau_{\bar{a},\bar{b}}(\bar{h}_t))^2] \lesssim r_{\mathrm{PO}}^{\bar{a},\bar{b}}(n) + \mathbb{E}\left[\hat{b}_{\mathrm{IPW}}^{\bar{a}}(\bar{h}_t)^2\right] + \mathbb{E}\left[\hat{b}_{\mathrm{IPW}}^{\bar{b}}(\bar{h}_t)^2\right] \tag{140}$$

$$\lesssim r_{\mathrm{PO}}^{\bar{a},\bar{b}}(n) + \max_{\ell \in \{t,\ldots,t+\tau\}} r_{\pi_\ell}^{\bar{a}}(n) + \max_{\ell \in \{t,\ldots,t+\tau\}} r_{\pi_\ell}^{\bar{b}}(n). \tag{141}$$

**DR learner.** We derive the bias term for the DR-learner via

$$\hat{b}_{\mathrm{DR}}^{\bar{a}}(\bar{h}_t) \tag{142}$$

$$= \mathbb{E}\left[\ldots \mathbb{E}\left[\left(\prod_{\ell=t}^{t+\tau} \frac{\mathbb{1}\{A_\ell = a_\ell\}}{\hat{\pi}_\ell(a_\ell \mid \bar{H}_\ell)} - \frac{\mathbb{1}\{A_\ell = a_\ell\}}{\pi_\ell(a_\ell \mid \bar{H}_\ell)}\right) Y_{t+\tau}\right.\right. \tag{143}$$

$$+ \sum_{k=t}^{t+\tau} \hat{\mu}_k^{\bar{a}}\left(\bar{H}_k\right)\left(1 - \frac{\mathbb{1}\{A_k = a_k\}}{\hat{\pi}_k(a_k \mid \bar{H}_k)}\right) \prod_{\ell=t}^{k-1} \frac{\mathbb{1}\{A_\ell = a_\ell\}}{\hat{\pi}_\ell(a_\ell \mid \bar{H}_\ell)} \tag{144}$$

$$\left.\left.- \sum_{k=t}^{t+\tau} \mu_k^{\bar{a}}\left(\bar{H}_k\right)\left(1 - \frac{\mathbb{1}\{A_k = a_k\}}{\pi_k(a_k \mid \bar{H}_k)}\right) \prod_{\ell=t}^{k-1} \frac{\mathbb{1}\{A_\ell = a_\ell\}}{\pi_\ell(a_\ell \mid \bar{H}_\ell)} \;\middle|\; \bar{H}_{t+\tau}\right]\ldots \middle| \bar{H}_t = \bar{h}_t\right] \tag{145}$$

$$\overset{(*)}{=} \mathbb{E}\left[\ldots \mathbb{E}\left[\sum_{k=t}^{t+\tau} \prod_{\ell=t}^{k-1} \frac{\mathbb{1}\{A_\ell = a_\ell\}}{\hat{\pi}_\ell(a_\ell \mid \bar{H}_\ell)} \frac{\pi_k(a_k \mid \bar{H}_k) - \hat{\pi}_k(a_k \mid \bar{H}_k)}{\hat{\pi}_k(a_k \mid \bar{H}_k)} \mu_k^{\bar{a}}\left(\bar{H}_k\right)\right.\right. \tag{146}$$

$$+ \sum_{k=t}^{t+\tau} \hat{\mu}_k^{\bar{a}}\left(\bar{H}_k\right)\left(1 - \frac{\mathbb{1}\{A_k = a_k\}}{\hat{\pi}_k(a_k \mid \bar{H}_k)}\right) \prod_{\ell=t}^{k-1} \frac{\mathbb{1}\{A_\ell = a_\ell\}}{\hat{\pi}_\ell(a_\ell \mid \bar{H}_\ell)} \tag{147}$$

$$\left.\left.- \sum_{k=t}^{t+\tau} \mu_k^{\bar{a}}\left(\bar{H}_k\right)\left(1 - \frac{\mathbb{1}\{A_k = a_k\}}{\pi_k(a_k \mid \bar{H}_k)}\right) \prod_{\ell=t}^{k-1} \frac{\mathbb{1}\{A_\ell = a_\ell\}}{\pi_\ell(a_\ell \mid \bar{H}_\ell)} \;\middle|\; \bar{H}_{t+\tau}\right]\ldots \middle| \bar{H}_t = \bar{h}_t\right] \tag{148}$$

$$= \mathbb{E}\left[\ldots \mathbb{E}\left[\prod_{\ell=t}^{t+\tau-1} \frac{\mathbb{1}\{A_\ell = a_\ell\}}{\hat{\pi}_\ell(a_\ell \mid \bar{H}_\ell)} \frac{\pi_{t+\tau}(a_{t+\tau} \mid \bar{H}_{t+\tau}) - \hat{\pi}_{t+\tau}(a_{t+\tau} \mid \bar{H}_{t+\tau})}{\hat{\pi}_{t+\tau}(a_{t+\tau} \mid \bar{H}_{t+\tau})} \mu_{t+\tau}^{\bar{a}}\left(\bar{H}_{t+\tau}\right)\right.\right. \tag{149}$$

$$+ \hat{\mu}_{t+\tau}^{\bar{a}}\left(\bar{H}_{t+\tau}\right)\left(1 - \frac{\pi_{t+\tau}(a_{t+\tau} \mid \bar{H}_{t+\tau})}{\hat{\pi}_{t+\tau}(a_{t+\tau} \mid \bar{H}_{t+\tau})}\right) \prod_{\ell=t}^{t+\tau-1} \frac{\mathbb{1}\{A_\ell = a_\ell\}}{\hat{\pi}_\ell(a_\ell \mid \bar{H}_\ell)} \tag{150}$$

$$-\mu_{t+\tau}^{\bar{a}}\left(\bar{H}_{t+\tau}\right)\left(1-\frac{\pi_{t+\tau}(a_{t+\tau}\mid\bar{H}_{t+\tau})}{\pi_{t+\tau}(a_{t+\tau}\mid\bar{H}_{t+\tau})}\right)\underbrace{\prod_{\ell=t}^{t+\tau-1}\frac{\mathbb{1}\{A_\ell=a_\ell\}}{\pi_\ell(a_\ell\mid\bar{H}_\ell)}}_{=0}\mid\bar{H}_{t+\tau-1}\Bigg]\cdots\Bigg|\bar{H}_t=\bar{h}_t\Bigg] \tag{151}$$

$$+\,\mathbb{E}\left[\ldots\mathbb{E}\left[\sum_{k=t}^{t+\tau-1}\prod_{\ell=t}^{k-1}\frac{\mathbb{1}\{A_\ell=a_\ell\}}{\hat{\pi}_\ell(a_\ell\mid\bar{H}_\ell)}\frac{\pi_k(a_k\mid\bar{H}_k)-\hat{\pi}_k(a_k\mid\bar{H}_k)}{\hat{\pi}_k(a_k\mid\bar{H}_k)}\mu_k^{\bar{a}}\left(\bar{H}_k\right)\right.\right. \tag{152}$$

$$+\sum_{k=t}^{t+\tau-1}\hat{\mu}_k^{\bar{a}}\left(\bar{H}_k\right)\left(1-\frac{\mathbb{1}\{A_k=a_k\}}{\hat{\pi}_k(a_k\mid\bar{H}_k)}\right)\prod_{\ell=t}^{k-1}\frac{\mathbb{1}\{A_\ell=a_\ell\}}{\hat{\pi}_\ell(a_\ell\mid\bar{H}_\ell)} \tag{153}$$

$$-\sum_{k=t}^{t+\tau-1}\mu_k^{\bar{a}}\left(\bar{H}_k\right)\left(1-\frac{\mathbb{1}\{A_k=a_k\}}{\pi_k(a_k\mid\bar{H}_k)}\right)\prod_{\ell=t}^{k-1}\frac{\mathbb{1}\{A_\ell=a_\ell\}}{\pi_\ell(a_\ell\mid\bar{H}_\ell)}\mid\bar{H}_{t+\tau-1}\Bigg]\cdots\Bigg|\bar{H}_t=\bar{h}_t\Bigg] \tag{154}$$

$$=\mathbb{E}\left[\prod_{\ell=t}^{t+\tau-1}\frac{\mathbb{1}\{A_\ell=a_\ell\}}{\hat{\pi}_\ell(a_\ell\mid\bar{H}_\ell)}\frac{\pi_{t+\tau}(a_{t+\tau}\mid\bar{H}_{t+\tau})-\hat{\pi}_{t+\tau}(a_{t+\tau}\mid\bar{H}_{t+\tau})}{\hat{\pi}_{t+\tau}(a_{t+\tau}\mid\bar{H}_{t+\tau})}\right. \tag{155}$$

$$\left.\left(\hat{\mu}_{t+\tau}^{\bar{a}}\left(\bar{H}_{t+\tau}\right)-\mu_{t+\tau}^{\bar{a}}\left(\bar{H}_{t+\tau}\right)\right)\mid\bar{H}_t=\bar{h}_t\right] \tag{156}$$

$$+\,\mathbb{E}\left[\ldots\mathbb{E}\left[\sum_{k=t}^{t+\tau-1}\prod_{\ell=t}^{k-1}\frac{\mathbb{1}\{A_\ell=a_\ell\}}{\hat{\pi}_\ell(a_\ell\mid\bar{H}_\ell)}\frac{\pi_k(a_k\mid\bar{H}_k)-\hat{\pi}_k(a_k\mid\bar{H}_k)}{\hat{\pi}_k(a_k\mid\bar{H}_k)}\mu_k^{\bar{a}}\left(\bar{H}_k\right)\right.\right. \tag{157}$$

$$+\sum_{k=t}^{t+\tau-1}\hat{\mu}_k^{\bar{a}}\left(\bar{H}_k\right)\left(1-\frac{\mathbb{1}\{A_k=a_k\}}{\hat{\pi}_k(a_k\mid\bar{H}_k)}\right)\prod_{\ell=t}^{k-1}\frac{\mathbb{1}\{A_\ell=a_\ell\}}{\hat{\pi}_\ell(a_\ell\mid\bar{H}_\ell)} \tag{158}$$

$$-\sum_{k=t}^{t+\tau-1}\mu_k^{\bar{a}}\left(\bar{H}_k\right)\left(1-\frac{\mathbb{1}\{A_k=a_k\}}{\pi_k(a_k\mid\bar{H}_k)}\right)\prod_{\ell=t}^{k-1}\frac{\mathbb{1}\{A_\ell=a_\ell\}}{\pi_\ell(a_\ell\mid\bar{H}_\ell)}\mid\bar{H}_{t+\tau-1}\Bigg]\cdots\Bigg|\bar{H}_t=\bar{h}_t\Bigg] \tag{159}$$

$$=\ldots \tag{160}$$

$$=\sum_{k=t}^{t+\tau}\mathbb{E}\left[\prod_{\ell=t}^{k-1}\frac{\mathbb{1}\{A_\ell=a_\ell\}}{\hat{\pi}_\ell(a_\ell\mid\bar{H}_\ell)}\frac{\pi_k(a_k\mid\bar{H}_k)-\hat{\pi}_k(a_k\mid\bar{H}_k)}{\hat{\pi}_k(a_k\mid\bar{H}_k)}\left(\hat{\mu}_k^{\bar{a}}\left(\bar{H}_k\right)-\mu_k^{\bar{a}}\left(\bar{H}_k\right)\right)\Bigg|\bar{H}_t=\bar{h}_t\right], \tag{161}$$

where $(\ast)$ follows from the bias formula of the IPW-learner.

Hence,

$$\mathbb{E}\left[\hat{b}_{\mathrm{DR}}^{\bar{a}}(\bar{h}_t)^2\right] \tag{162}$$

$$\overset{(\ast)}{\le}C\,\mathbb{E}\left[\sum_{k=t}^{t+\tau}\mathbb{E}\left[\left(\pi_k(a_k\mid\bar{H}_k)-\hat{\pi}_k(a_k\mid\bar{H}_k)\right)^2\left(\hat{\mu}_k^{\bar{a}}\left(\bar{H}_k\right)-\mu_k^{\bar{a}}\left(\bar{H}_k\right)\right)^2\mid\bar{H}_t=\bar{h}_t\right]\right] \tag{163}$$

$$\overset{(\ast\ast)}{=}C\sum_{k=t}^{t+\tau}\mathbb{E}\left[\mathbb{E}\left[\left(\pi_k(a_k\mid\bar{H}_k)-\hat{\pi}_k(a_k\mid\bar{H}_k)\right)^2\right]\mathbb{E}\left[\left(\hat{\mu}_k^{\bar{a}}\left(\bar{H}_k\right)-\mu_k^{\bar{a}}\left(\bar{H}_k\right)\right)^2\right]\mid\bar{H}_t=\bar{h}_t\right] \tag{164}$$

$$\lesssim\sum_{\ell=t}^{t+\tau}r_{\pi_\ell}^{\bar{a}}(n)\left(\sum_{k=\ell}^{t+\tau}r_{\mu_k}^{\bar{a}}(n)\right), \tag{165}$$

where $(\ast)$ follows from the boundedness assumption (2) and $(\ast\ast)$ follows from Fubini's theorem and the sample splitting assumption (4). For CATE, we obtain the corresponding rate via

$$\mathbb{E}[(\hat{\tau}_{\mathrm{DR}}^{\bar{a},\bar{b}}(\bar{h}_t)-\tau_{\bar{a},\bar{b}}(\bar{h}_t))^2]\lesssim r_{\mathrm{PO}}^{\bar{a},\bar{b}}(n)+\mathbb{E}\left[\hat{b}_{\mathrm{DR}}^{\bar{a}}(\bar{h}_t)^2\right]+\mathbb{E}\left[\hat{b}_{\mathrm{DR}}^{\bar{b}}(\bar{h}_t)^2\right] \tag{166}$$

$$\lesssim r_{\mathrm{PO}}^{\bar{a},\bar{b}}(n)+\sum_{\ell=t}^{t+\tau}\left(r_{\pi_\ell}^{\bar{a}}(n)\sum_{k=\ell}^{t+\tau}r_{\mu_k}^{\bar{a}}(n)+r_{\pi_\ell}^{\bar{b}}(n)\sum_{k=\ell}^{t+\tau}r_{\mu_k}^{\bar{b}}(n)\right). \tag{167}$$

□

# D    MATHEMATICAL DETAILS REGARDING ASSUMPTION 3

In this section, we provide background on the formal definitions of the convergence rates in Assumption 3. We follow Stone (1980). Let $\theta$ be a parameter and $T(\theta)$ some target functional that we want to estimate.

**Definition 1.** A sequence of estimators $\hat{T}_n(\theta)$ of a functional $T(\theta)$ converges with (achievable) rate $r_T(n)$ if

$$\lim_{c \to 0} \liminf_{n \in \mathbb{N}} \sup_{\theta \in \Theta} \mathbb{P}_\theta(\hat{T}_n(\theta) - T(\theta)| > cr_\theta(n)) = 0. \tag{168}$$

**Definition 2.** A rate $r_T(n)$ is called an upper bound to the rate of convergence if for all estimators $\hat{T}_n(\theta)$ it holds for all $c > 0$ that

$$\liminf_{n \in \mathbb{N}} \sup_{\theta \in \Theta} \mathbb{P}_\theta(|\hat{T}_n(\theta) - T(\theta)|| > cr_\theta(n)) > 0 \tag{169}$$

and

$$\lim_{c \to 0} \liminf_{n \in \mathbb{N}} \sup_{\theta \in \Theta} \mathbb{P}_\theta(\hat{T}_n(\theta) - T(\theta)| > cr_\theta(n)) = 1. \tag{170}$$

$r_T(n)$ is called optimal if it is both achievable and an upper bound.

Optimal rates are known in a variety of settings and depend on the specific assumption of the underlying function class. In the following, we provide two examples of optimal rates that could be used in our Theorem 1 to obtain explicit rates if we impose the corresponding assumptions on our nuisance functions and second-stage regression functions (similar as in, e.g., Curth & van der Schaar (2021); Kennedy (2023)).

**Smoothness assumptions:** Stone (1980) showed that for a nonparametric regression problem with an $\eta$-smooth regression function, the optimal rate of convergence is $n^{-\frac{2\eta}{2\eta+p}}$, where $p$ is the dimension of the covariate space.

**Sparsity assumptions:** We say that a function $f(x)$ is $k$-sparse, if it is linear in $x \in \mathbb{R}^p$ and it only depends on $k < \min\{n, p\}$ predictors. (Yang & Tokdar, 2015) showed, that in this case the optimal rate of the regression problem is given by $\frac{k \log(p)}{n}$. The linearity assumption can be relaxed to an additive structural assumption. Sparsity assumptions are often imposed in high-dimensional settings where $n < p$.

# E  IMPLEMENTATION AND HYPERPARAMETER DETAILS

**General implementation of nuisance and second-stage regressions:** We start by noting that each nuisance and second-stage estimator can be written as a function $f \colon \mathcal{H} \to \mathbb{R}$ that uses the history $\bar{H}_t$ to predict a $\kappa$-step ahead target $\widetilde{Y}_{t+\kappa}$. For example, for the PI-HA-learner, we set $\widetilde{Y}_{t+\tau} = Y_{t+\tau}$. We parametrize each learner via $f_\theta(\bar{h}_t) = g_\eta(\Phi_\nu(\bar{h}_t))$, where $\theta = (\eta, \nu)$ are learnable parameters, $\Phi_\nu \colon \mathcal{H} \to \mathbb{R}^p$ is a parametrized representation function, and $g_\eta \colon \mathbb{R}^p \to \mathbb{R}$ is a parametrized output function. Finally, we learn the propensity scores $\pi_\ell(a_\ell \mid \cdot)$ in a joint architecture, while using separate models for the response functions $\mu_\ell^{\bar{a}}(\cdot)$ (as they require nested regressions).

**Transformer architecture.** We build on an encoder transformer as in (Ashish Vaswani et al., 2017). Each transformer consists of a single transformer block and employs a causal mask to avoid look-ahead bias. First, the input is passed through a linear input layer. In order to ensure that the sequence order is preserved, we apply a non-trainable positional encoding. Then, the history is passed through the transformer blocks. Each block consists of (i) a self-attention mechanism with three attention heads and hidden state dimension $d_{\text{model}} = 30$, (ii) and a feed-forward network with hidden layer size $d_{\text{ff}} = 20$. Both the (i) self-attention mechanism and (ii) the feed-forward network employ residual connections, which are followed by dropout layers with dropout probabilities $p = 0.1$, respectively. For regularization, we apply post-normalization as in (Ashish Vaswani et al., 2017), that is, layer normalization after each residual connection. Finally, we send the learned representation through a feed-forward neural network with one hidden layer of size $d'_{\text{ff}} = 20$, ReLU nonlinearities, and with linear or softmax activation for regression and classification tasks, respectively.

**Training:** We perform training by minimizing the loss $\mathcal{L}(\theta) = \sum_{i=1}^{n} \sum_{t=1}^{T^{(i)}-\kappa} \frac{w_t^{(i)}}{(T^{(i)}-\kappa)} \left( \widetilde{y}_{t+\tau}^{(i)} - f_\theta\left(\bar{h}_t^{(i)}\right) \right)^2$ where $w_t^{(i)}$ are either observations of the stabilized inverse-variance weights from Eq. (20) or $w_t^{(i)} = 1/n$ for all $i$ and $t$. If the target $\widetilde{Y}_{t+\tau}$ is discrete (i.e., for learning propensity scores), we use a cross-entropy instead of mean squared error. For nuisance estimators which additionally condition on treatments (e.g., history adjustments or response functions), we split the data and fit the estimator on the part corresponding to the treatment intervention (as done by the T-learner (Curth et al., 2020) in the static setting). Note that, in our loss, we not only average over samples but also over time steps. This is standard in the time-varying setting and increases the effective sample size by leveraging the time dimension (Lim et al., 2018; Bica et al., 2020a).

**Hyperparameters.** To ensure a fair comparison, we choose the same hyperparameters (e.g., dimensions, learning rate) for all nuisance estimators and meta-learners. We employ additional weight decay for the two-stage learners to avoid overfitting during the pseudo-outcome regression (as the CATE is of simpler structure than the nuisance parameters). This is consistent with the literature on meta-learners for CATE estimation (Curth et al., 2020). For reproducibility purposes, we report all hyperparameters as `.yaml` files.[3]

**Runtime.** For each transformer-based learner, training took approximately 90 seconds using $n = 5000$ samples and a standard computer with AMD Ryzen 7 Pro CPU and 32GB of RAM.

---

[3]Code is available at https://github.com/DennisFrauen/CATEMetaLearnersTime.

## F    REMARKS ON BALANCED REPRESENTATIONS

Prior works CAPO/ CATE estimation propose to learn balanced representations that are non-predictive of the treatment (Bica et al., 2020b; Melnychuk et al., 2022; Seedat et al., 2022). The idea is to mimic randomized clinical trials and reduce finite-sample error due to estimation variance (Shalit et al., 2017). However, for time-varying treatment effects, this approach has several drawbacks:

1. Learning guarantees only exist in static (i.e., non-time-varying) settings (Shalit et al., 2017). Thus, balanced representations do not resolve bias due to time-varying confounders. In particular, all mentioned works (Bica et al., 2020b; Melnychuk et al., 2022; Seedat et al., 2022) can be interpreted as PI-HA-learners with an additional balancing loss component. As we have shown, special adjustment methods are necessary to remove bias from time-varying confounders, as done by the PI-RA, RA-, IPW-, DR-, and IPW-DR-learner.

2. Even in static settings, methods based on balanced representations impose invertibility assumptions on the learned representations (Shalit et al., 2017). This is often unrealistic in time-varying settings, where representations not only incorporate information about the current confounders but also about the full patient history.

3. Because invertibility is difficult to ensure, balanced representations may increase bias (a phenomenon called representation-induced confounding (Melnychuk et al., 2024). We refer to Curth et al. (2021) and Melnychuk et al. (2024) for a detailed discussion on this issue. Again, this is particularly detrimental in time-varying settings.

For the above reasons, we decided not to incorporate balanced representations into our model instantiations when running our experiments. This is consistent with previous works in the literature (Curth et al., 2021; Vanderschueren et al., 2023). Nevertheless, we emphasize that balanced representations can be easily integrated into our meta-learners.

## G  DETAILS REGARDING EXPERIMENTS ON SIMULATED DATA

**Data-generating process.** Our general data-generating process for simulating datasets is as follows: we start by simulating an initial confounder $X_1 \sim \mathcal{N}(0,1)$ from a standard normal distribution. For the following time steps $t \in \{2, \ldots, 5\}$, we simulate time-varying confounders via

$$X_t = f_x(Y_{t-1}, A_{t-1}, \bar{H}_{t-1}) + \varepsilon_x, \tag{171}$$

for some function $f_x$ and where $\varepsilon_x \sim \mathcal{N}(0, 0.5)$. Then, we simulate binary, time-varying treatments via

$$A_t = x \sim \text{Bernoulli}\,(p_t) \quad \text{with} \quad p_t = \sigma(f_a(\bar{H}_t)) \tag{172}$$

for some function $f_a$ and where $\sigma(\cdot)$ denotes the sigmoid function. Finally, we simulate time-varying, continuous outcomes via

$$Y = f_y(A_t, \bar{H}_t) + \varepsilon_y, \tag{173}$$

where $\varepsilon_y \sim \mathcal{N}(0, 0.3)$.

• **Dataset $\mathcal{D}_1$.** For $\mathcal{D}_1$, we set the confounding function to $f_x(Y_{t-1}, A_{t-1}, \bar{H}_{t-1}) = 0.5X_{t-1}$, the treatment assignment function to $f_a(\bar{H}_t) = 4\cos(0.5X_t - 0.5(A_{t-1} - 0.5))$, and the outcome assignment function to $f_y(A_t, \bar{H}_t) = \cos(X_t) + 0.5(A_t - 0.5)$. We sample a training dataset of size $n_{\text{train}} = 5000$ and a test dataset of size $n_{\text{test}} = 1000$.

• **Dataset $\mathcal{D}_2$.** For $\mathcal{D}_2$, we set the confounding function to $f_x(Y_{t-1}, A_{t-1}, \bar{H}_{t-1}) = 0.5X_{t-1}$, the treatment assignment function to $f_a(\bar{H}_t) = 0.5X_t - 0.5(A_{t-1} - 0.5)$, and the outcome assignment function to $f_y(A_t, \bar{H}_t) = \cos(5X_t) + 0.5(A_t - 0.5)$. We sample a training dataset of size $n_{\text{train}} = 10000$ and a test dataset of size $n_{\text{test}} = 1000$.

• **Dataset $\mathcal{D}_3$.** For $\mathcal{D}_3$, we set the confounding function to $f_x(Y_{t-1}, A_{t-1}, \bar{H}_{t-1}) = 0.5X_{t-1}$, the treatment assignment function to $f_a(\bar{H}_t) = \gamma(0.5X_t - 0.5(A_{t-1} - 0.5))$, where $\gamma$ is a parameter controlling the overlap, and the outcome assignment function to $f_y(A_t, \bar{H}_t) = \cos(X_t) + 0.5(A_t - 0.5)$. We sample a training dataset of size $n_{\text{train}} = 5000$ and a test dataset of size $n_{\text{test}} = 1000$.

**Interventions.** For $\mathcal{D}_1$ and $\mathcal{D}_2$ we set $\tau \in \{0, 1, 2\}$ and choose the corresponding treatment interventions $a_\tau$ and $b_\tau$ as follows: $a_0 = 1$, $a_1 = (0,1)$, $a_2 = (0,0,1)$, $b_0 = 0$, $b_1 = (1,0)$, and $b_2 = (1,0,0)$. Hence, the corresponding CATE measures the effect of intervening at the last time step as compared to intervening at the first time step. For $\mathcal{D}_2$, we set $\tau = 1$, $a_1 = (0,1)$, and $b_1 = (1,0)$.

# H ADDITIONAL EXPERIMENTS USING LSTM-BASED MODEL ARCHITECTURES

Our meta-learners are model agnostic and can thus be instantiated with any machine learning model. Here, we perform additional experiments for an instantiation using LSTM networks (Hochreiter & Schmidhuber, 1997). The results are shown in Tables 6 and 7. The results remain largely consistent with the ones from our transformer-based instantiations.

Table 6: Results for $\mathcal{D}_1$.

|  | $\tau = 0$ | $\tau = 1$ | $\tau = 2$ |
|---|---|---|---|
| PI-HA-learner | $1.99 \pm 0.15$ | $8.37 \pm 0.34$ | $8.83 \pm 0.44$ |
| PI-RA-learner | $2.00 \pm 0.16$ | $5.94 \pm 0.21$ | $3.58 \pm 0.35$ |
| RA-learner | $0.05 \pm 0.04$ | $\mathbf{0.13 \pm 0.10}$ | $0.68 \pm 0.42$ |
| IPW-learner | $0.10 \pm 0.06$ | $0.20 \pm 0.11$ | $\mathbf{0.32 \pm 0.16}$ |
| DR-learner | $0.09 \pm 0.04$ | $0.17 \pm 0.13$ | $0.35 \pm 0.21$ |
| IVW-DR-learner | $\mathbf{0.05 \pm 0.02}$ | $0.17 \pm 0.11$ | $0.66 \pm 0.31$ |

Results averaged over 5 random seeds. Best learner in bold.

Table 7: Results for $\mathcal{D}_2$.

|  | $\tau = 0$ | $\tau = 1$ | $\tau = 2$ |
|---|---|---|---|
| PI-HA-learner | $3.23 \pm 0.61$ | $14.89 \pm 1.92$ | $19.43 \pm 2.25$ |
| PI-RA-learner | $2.83 \pm 0.20$ | $6.53 \pm 0.43$ | $7.42 \pm 0.64$ |
| RA-learner | $0.20 \pm 0.18$ | $1.07 \pm 0.64$ | $4.94 \pm 1.03$ |
| IPW-learner | $0.21 \pm 0.16$ | $0.80 \pm 0.61$ | $6.96 \pm 2.64$ |
| DR-learner | $0.20 \pm 0.13$ | $\mathbf{0.72 \pm 0.44}$ | $7.33 \pm 2.00$ |
| IVW-DR-learner | $\mathbf{0.13 \pm 0.11}$ | $0.95 \pm 0.46$ | $\mathbf{4.33 \pm 1.27}$ |

Results averaged over 5 random seeds. Best learner in bold.

# I    Additional experiments using real-world data

We provide an additional exemplary application of our meta-learners for a specific patient from the MIMIC data who was never treated (Fig. 4). The counterfactual outcomes predicted by the meta-learners for $t \geq 6$ indicate that providing the patient with ventilation would have decreased their expected blood pressure for future time steps. While evaluating the best counterfactual trajectories is not possible on real-world data, the results may indicate that PIHA and PIRA overestimate the treatment effect while IPW and DR underestimate the effect. In contrast, our IVW-DR-learner provides a balanced trajectory.

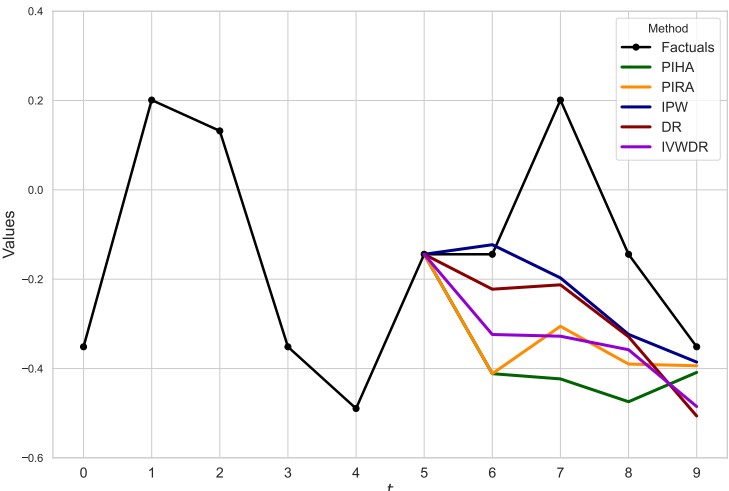

Figure 4: **Results for real-world medical data**. Blood pressure predictions of meta-learners for a single patient that was never treated.

