# OpenReview forum: "Model-agnostic meta-learners for estimating heterogeneous treatment effects over time"
_ICLR.cc/2025/Conference — ICLR 2025 Poster_

### Official Review · Reviewer_spHK · 2024-10-17

**Soundness:** 4
**Presentation:** 3
**Contribution:** 3
**Rating:** 8
**Confidence:** 3

**Summary:**

The authors compare different causal metalearners estimating treatment outcomes over time. They also propose a novel variant of the temporal DR-Learner based on inverse-variance weights. These learners are described and then compared based on a transformer architecture.

**Strengths:**

The presented metalearners are a novel framework for thinking about these models and, as such, this work constitutes an important contribution to the literature. Given the popularity of metalearners in the static setting, these will certainly have wide applicability. Moreover, given the fast evolution in research on time series modeling, having these model-agnostic metalearners will be very valuable. The presented experiments clearly show the benefits of different approaches, with results corresponding to the hypotheses suggested in the theoretical analysis.

**Weaknesses:**

1. Several points are not entirely clear to me (possibly due to some misunderstandings):
- If I understand correctly, the challenge of time-varying confounding is different compared to previous work (e.g., CRN). In my opinion, this needs to be stressed more and explained in more detail.

- Some parts of the methodology are not entirely clear (see questions).

- The notation is not always clear to me (see questions).


2. There is no conclusion, or a discussion of limitations and possible directions for future work.

**Questions:**

- I am not sure whether I understand the implications of eq. 3. The authors claim that other methods, such as CRN and CT, do not correctly adjust for this bias. However, as they build an unbiased representation iteratively in each step (based on $\bar{H}_t$), it seems to me that future covariates get added correctly, and no bias should exist?

- Do the learners with different estimands (e.g., nuisance parameters) use separate models for each, or are they built in a multitask framework? While a multitask framework is sensible for transformers, it does not apply to time series models generally.

- Do you use sample splitting in the experiments?

- Having CAPO as $\mu$ and CATE as $\tau$ would be more clear in my opinion, though I leave this up to the authors. Later, $\mu$ and $\delta$ are also introduced, though they seemingly refer to similar estimands. What is the reasoning behind this notation?

- The RA-Learner is much more complex than its static variant, where only the final term in Eq. 8 is present. Some intuition here would be appreciated.

___

**Minor points:**
- Line 290: "major different" -> difference

---

> ### Author Response · Authors · 2024-11-17
> **Response to Reviewer spHK**
>
> Thank you for your detailed feedback and insightful questions. We appreciate the opportunity to address your concerns and provide clarifications. Below, we respond to the specific weaknesses and questions raised in your review:
>
>
> ## Response to “Weaknesses”
>
>
>
> 1. **Clarifications and improvements:** Thank you for highlighting aspects of our work that needed clarification. To avoid redundancy, we refer to our responses to the questions below. **Action:** We incorporated all clarifications directly into the revised paper.
> 2. **New discussion section:** We appreciate your suggestion to expand our discussion and add a conclusion section. **Action:** We added a dedicated discussion section that includes a summary of key contributions, a discussion of limitations and their implications, and directions for future research.
>
>
> ### Response to “Questions”
>
>
>
> 1. **Implications of Eq. (3) and time-varying confounding:** Thank you for this excellent question regarding bias due to time-varying confounding and methods like CRN.
>     * Eq. (3) states that methods relying solely on the current history and future treatments to predict the $\tau$-step-ahead outcome will be biased for estimating CAPO/CATE. Intuitively, **these methods ignore future covariates occurring after/during the treatment intervention**, which act as "post-treatment confounders." If these covariates are omitted during training, they essentially become "unobserved confounders," introducing bias. Note that we are not the first to obtain this insight; this is a well-known fact in the causal inference literature [e.g., 1]. For unbiased estimation, proper time-varying adjustment mechanisms (e.g., iteratively integrating out post-treatment covariates as shown in Eq. (5) and Eq. (6)) are required.
>     * **Regarding CRN**: The CRN can be interpreted as an **instantiation of the PI-HA learner**, using an encoder-decoder LSTM architecture. It learns a representation of the current history (encoder) and predicts $\tau$-step-ahead outcomes using this representation and the intervention sequence (decoder). However, because the CRN ignores post-treatment confounders, it is inherently biased and does not converge to the true CAPO/CATE, even with infinite data. Importantly, the PI-HA learner (and thus CRN) may still outperform other learners in practice due to trade-offs between asymptotic bias and finite-sample variance, which depend on factors such as sample size, time-varying confounding strength, and data complexity.
>     * **Regarding balancing:** Balancing modifies the loss to learn a representation that is independent of the treatment (motivated by randomized trials). However, balancing does **not** address bias due to time-varying confounding as it ignores the post-treatment confounders. Instead, balancing aims to reduce **finite sample variance** (see [2]) and thus does not affect our **asymptotic analysis** (Theorem 1). Furthermore, balancing has the following drawbacks for the time-series setting: (i) guarantees currently only exist for the static setting [2]; (ii) balancing requires invertible representations, which is hard to fulfill in the time-series setting; and (iii) recent work has shown that balancing can even induce bias if representations are not invertible, so-called representation-induced confounding bias [3].
>     * **Action:** We added **a new Appendix F** to discuss connections between time-varying confounding, asymptotic bias, finite-sample variance, and balancing.
> 2. **Separate models vs. multitask framework:**
>     * In our work, we use separate models for each nuisance parameter. However, we acknowledge that a multitask framework could be applied, particularly for architectures like Transformers. Such a framework may perform better if the propensity scores and response functions share structural similarities (or worse if they don’t), as noted in [4, 5].
>     * Our paper focuses on the theoretical and empirical properties of meta-learners rather than optimizing underlying nuisance models. Exploring multitask frameworks is an exciting direction for future work.
>     * **Action:** We added a discussion on multitask frameworks and mentioned their potential benefits, which we added as a new research outlook to our revised paper.

---

> > ### Author Response · Authors · 2024-11-17
> > **Response to Reviewer spHK #2**
> >
> > 3. **Sample splitting:** In our experiments, we did not use sample splitting for the following reasons:
> >     * Sample splitting is **primarily a tool to simplify theoretical analysis** in semiparametric statistics by controlling the remainder of the empirical process term (we refer to [6] for a nice overview). Intuitively, it prevents overfitting by using the same data twice for nuisance and second-stage estimation. However, sample splitting is not the only way of ensuring this. For example, the theory still works if nuisance models lie within specific function classes (e.g., Donsker classes [6]), which include all smooth parametric models (e.g., neural networks with fixed architectures) [7]. In our paper, we prevented the models from overfitting by monitoring performance on a separate validation dataset.
> >     * Sample splitting can harm finite-sample performance by reducing the data available for training, which is why prior works often avoided it in experiments (e.g., [4]).
> >     * Our experimental results align with theoretical expectations even without using sample splitting, making its inclusion unnecessary.
> >     * **Action:** We added a clarification on sample splitting to the revised paper. We are open to including it if you prefer.
> > 4. **Notation Clarity:** We apologize for any confusion caused by our notation. **Action:**
> >     * We followed your suggestion and changed our notation for the CAPO from $\tau$ to $\mu$.
> >     * $\delta$ represents the $\tau$-step-ahead conditional expectation based on the history at time $t$, used solely for the PI-HA learner. We are happy to change the symbol if you prefer.
> > 5. **RA-Learner Complexity:** The RA-learner is more complex than its static counterpart due to:
> >     * The final term in Eq. (8) accounts for the possibility of categorical treatments and ensures unbiasedness.
> >     * We use the next time step’s response function in the pseudo-outcome instead of the observed outcome because this ensures the unbiasedness of the pseudo-outcome when being used as a second-stage regression target.
> >     * **Action:** We have added **additional proofs** in our **new Appendix C.2**, explicitly showing that all pseudo-outcomes (including RA) considered in our paper are unbiased and can be used as second-stage regression targets.
> >
> >
> > ### Response to “Minor point”
> >
> > Thank you for pointing this out. **Action:** We corrected the typo in Line 290 and revised other minor grammatical issues in the paper.
> >
> > We hope these responses address your concerns and improve the clarity of our work. Thank you again for your thoughtful feedback.
> >
> >
> > ## References
> >
> > [1] van der Laan, Gruber (2012). Targeted Minimum Loss Based Estimation of Causal Effects of Multiple Time Point Interventions. The International Journal of Biostatistics.
> >
> > [2] Shalit et al. (2017). Estimating individual treatment effect: generalization bounds and algorithms. ICML.
> >
> > [3] Melnychuk et al. (2024). Bounds on Representation-Induced Confounding Bias for Treatment Effect Estimation. ICLR.
> >
> > [4] Curth, van der Schaar (2021). Nonparametric estimation of heterogeneous treatment effects: From theory to learning algorithms. AISTATS.
> >
> > [5] Curth, van der Schaar (2021). On Inductive Biases for Heterogeneous Treatment Effect Estimation. NeurIPS.
> >
> > [6] Kennedy (2022). Semiparametric doubly robust targeted double machine learning: a review. Handbook of Statistical Methods for Precision Medicine.
> >
> > [7] Van der Vaart (2000). Asymptotic statistics. Cambridge University Press.

---

> > > ### Comment · Reviewer_spHK · 2024-11-18
> > >
> > > Thank you very much for the detailed response! My concerns have been addressed, and I have updated my score accordingly. Great work!

---

> > > > ### Author Response · Authors · 2024-11-18
> > > > **Thank you**
> > > >
> > > > Thank you for your fast response to our rebuttal and for increasing your score. As promised, we will include all changes in the camera-ready version of our paper.

---

### Official Review · Reviewer_Mx4T · 2024-10-30

**Soundness:** 2
**Presentation:** 2
**Contribution:** 2
**Rating:** 6
**Confidence:** 3

**Summary:**

This paper studies treatment effect estimation from observational data over a period of time. The observed data contains a (time-varying) covariate, treatment, and outcome at each time point. The proposed methods contain a set of meta-learners designed to be used with any machine learning algorithm as base learners. The meta-learners are specifically designed for four types of confounding adjustment, namely history adjustment, regression adjustment, propensity adjustment, and doubly robust adjustment. In other words, the paper extends existing adjustment methods under the static setting into the dynamic setting. The theoretical bound between each of the proposed estimators and the true effect is given under assumptions that are commonly used in treatment effect estimation. In the experiment section, the paper studied Transformer as one type of learner under one synthetic and the MIMICIII dataset.

**Strengths:**

This paper's strength is that it provides a good organisation of the existing learners and discusses the meta-learners and their estimation bounds. Although the technique used to prove the bound is not novel, it is nice to see the combination of existing techniques to provide bounds for each of the meta-learner.

**Weaknesses:**

1. One key weakness is that the experiment section is weak. Currently, the content has only one synthetic dataset and one real-world dataset, and the compared methods only include the meta-learners proposed in this paper. Only one instantiation is experimentally studied for the meta-learners proposed in this paper. I think there should at least be some existing state-of-the-art comparison included, and there should also be other instantiations even for a "proof-of-demonstration" experiment section, especially since ICLR has a tradition of valuing comprehensive experiments.

2. The later sections of the paper are not as well-written as the earlier ones. The latter sections seem rushed, and many sentences are incorrect. For example, "Here, we section outline..." should be this section outline; "proof-of-demonstration" should be proof-of-concept; Making things worse, the paper ended abruptly at section 7. There is not even a conclusion paragraph at the end of the paper.

**Questions:**

1. One strength claimed by the author is that the proposed learners can be used with "arbitrary machine learning models", and the title also emphasises "model-agnostic meta-learners". However, this is not supported by the empirical evaluation at all. I suggest the authors back up their claim with evidence. Also, some comparisons with different machine-learning models are essential for readers to understand the proposed meta-learner better. In its current form, the paper feels very unfinished.

2. I think the paper is relatively well-written until section 7. Unfortunately, the rest of the paper seems very rushed, and the content is limited. In the only instantiation using a Transformer, the authors emphasized that the Transformer has not been optimized for the treatment effect estimation and then use quite a bit of space to give the details of their architecture. I think the space can be better used if more experiment results can be presented. The detailed architecture of Transformer can be in the supplementary.

3. The discussion of results should also be enhanced. Currently, there is only a very brief description of the results of the experiment. There should at least be discussion of the results that can be linked back to the theoretical analysis. For example, in the theorem, it has been shown that the PI-HA learner is asymptotically biased. Still, in the real-world dataset, there are two occasions that PI-HA performs the best and by a significant margin. In fact, except for the IVW-DR-learner, other proposed meta-learners seldom perform better than the PI-HA learner despite the fact that PI-HA is biased while the others are not.

---

> ### Author Response · Authors · 2024-11-17
> **Response to Reviewer Mx4T**
>
> Thank you for your constructive feedback and suggestions. We appreciate the opportunity to address your concerns and improve our paper. Below, we respond to the key points you raised:
>
>
> ## Response to “Weaknesses”
>
> 1. **Experiments:**
>     * **Model-agnostic property and additional results:**
>         * We kindly point you to **Appendix H**, where we **repeated experiments using a different model architecture** (LSTM instead of transformer). We apologize that this was not visible enough in our main paper. These results empirically demonstrate that our meta-learners are flexible with respect to the choice of the underlying model. **Action:** In our revised paper, we provided a more visible reference to Appendix F.
>         * Additionally, we would like to make the point that the "model-agnostic" property of our meta-learners **does not even require empirical justification or proof**. This property follows directly from the fact that our meta-learners are defined in terms of conditional expectations and probabilities. Estimating a conditional expectation, $\mathbb{Ε}[Y | X]$, is equivalent to minimizing the squared loss $\mathcal{L}(f) = \mathbb{E}[(Y - f(X))^2]$ over a sufficiently rich function class $f \in \mathcal{F}$. **This equivalence is well-established** in statistical theory and applies to any (machine learning) model class $\mathcal{F}$. **Action:** In our revised apper we elaborated on the term “model-agnostic”.
>     * **Choice of model for experiments:**
>         * Our experimental setup follows **standard practices in the causal inference literature**, where a **fixed model architecture is used across all meta-learners**. This approach ensures that performance differences can be attributed to the choice of meta-learners rather than the underlying models. Note that meta-learners have been studied already in a variety of other causal inference settings [e.g., 1, 2, 3, 4, 5, 6], and **our paper follows a well-established way of setting up experiments** in this community.
>         * We deliberately chose a standard Transformer architecture for two reasons:
>             1. A simpler architecture allows us to **isolate and evaluate the properties of the meta-learners themselves**.
>             2. Introducing a more sophisticated architecture for one meta-learner (e.g., the causal transformer [7] as an instantiation of the PIHA learner) would necessitate equivalent architectures for all meta-learners to ensure a fair comparison. However, it is **not fully clear what such architectures would look like** e.g., for the DR-learner. We agree that optimizing model architectures for specific meta-learners is an interesting direction for future work but goes beyond the scope of this work.
>         * In summary, **our focus is on the theoretical underpinnings of meta-learners and providing practical insights** (e.g., on what meta-learners to choose in which situations). Exploring more sophisticated model choices combined with our meta-learners is an interesting direction for future research but falls outside the scope of this paper.
>     * **Datasets:**
>         * We would like to clarify that we use **multiple datasets** for our evaluation, each with certain characteristics to show specific aspects of our theoretical results. While It is true that we only use one real-world dataset for our evaluation, we do this on purpose and **follow established literature** [e.g., 1, 2, 3, 4]. The reason is that **proper evaluation of causal inference methods using real-world datasets is impossible** due to missing causal ground truth. We thus follow established literature and report factual RMSE as a proxy for performance on real-world data, but focus on synthetic data for the main part of our experiments.
>     * **Action:** In our revised paper, we now elaborate more clearly on the way we set up our experiments.
> 2. **Improvements to later sections of the paper:**
>     * We appreciate your feedback on the later sections of the paper. **Action**: We have revised these according to your suggestions. Specifically:
>         * **Typos and grammatical errors:** We have corrected the errors you identified.
>         * **Model Description:** We followed your recommendation and moved parts of the detailed Transformer architecture description to the appendix.
>         * **Discussion and conclusion:** We added a discussion section, which includes a summary of key findings, limitations, and an outlook on future work.
>         * **Experimental results:** We expanded the discussion of our results, thereby adding more interpretations and linking them back to the theoretical analysis. For example, we now explain why the PI-HA learner performs well in certain scenarios despite being asymptotically biased (see also our response to question 3).

---

> > ### Author Response · Authors · 2024-11-17
> > **Response to Reviewer Mx4T #2**
> >
> > ## Response to “Questions”
> >
> > 1. **Empirical validation of model-agnostic property:**
> >     * We would again like to direct your attention to **Appendix H**, where we provide **additional results** for an **LSTM-based instantiation of our meta-learners**. These results illustrate the flexibility of our framework.
> >     * However, as noted above, the model-agnostic property of our meta-learners does not require empirical validation. It is a direct consequence of the equivalence between estimating conditional expectations and minimizing squared loss. **Action:** We explicitly highlighted the LSTM results in the main paper to ensure better visibility and elaborated on the term “model-agnostic”.
> > 2. **Transformer architecture details:**
> >     * We appreciate your suggestion to streamline the discussion of our Transformer-based implementation. **Action:** We moved much of the detailed architecture description in Section 7.1 to the appendix and used the freed space to elaborate on our experimental results and add a discussion (see our response 2 to weaknesses).
> > 3. **Interpretation of experimental results:**
> >     * **Why does the PI-HA learner perform well in Table 4 despite being biased?** We assume you are referring to the **results on real-world data** here (Table 4), since, in the synthetic experiments (Table 2+3), the PI-HA learner generally performs worse by a large margin than the other meta-learners. You are correct that we showed in Theorem 1 that the PI-HA learner is (asymptotically) biased. However, for the actual performance on (finite) data, a variety of other factors may influence the results:
> >         1. **Low overlap**. The real-world data we use has **low-overlap regions** (i.e., regions in which propensity scores are extreme -> close to zero or one). Low overlap regions especially affect the finite-sample performance of the IPW- and the DR-learner because these meta-learners divide by propensity scores (and thus effectively divide by small numbers close to zero, blowing up the variance). The remaining meta-learners do not suffer as strongly: PI-HA and PI-RA do not divide by propensities, and our IVW-DR-learner downweights such extreme observations by using our inverse variance weights from Theorem 2. This becomes even more severe for larger time horizons (products of propensities) **and is evident in Table 4**.
> >         2. **Time-varying confounding:** If there is no strong time-varying confounding (e.g. if the time-varying covariates $X_t$ do not have a strong effect on the treatments $A_t$), the bias of the PI-HA-learner is less severe. The time structure may be ignored and static meta-learners may provide a sufficient approximation.
> >         3. **Finite sample variance**. Specific meta-learners such as the PI-RA learner need to fit multiple models (e.g., one for each time step). This can lead to a larger finite sample variance, which is not accounted for in our (asymptotic) theoretical analysis in Theorem 1. Nevertheless, asymptotic analysis is standard in the literature and provides valuable insights for large sample sizes.
> >         4. **Evaluation using real-world data:** Evaluation of causal inference method using real-world data is notoriously hard due to missing causal ground truth. This is why followed established literature and used synthetic data for the main parts of our experiments (see also our response to 1 to weaknesses).
> >
> >         In sum, the **main takeaway** of Table 4 is that **our inverse variance weights are effective in stabilizing the DR-loss** even under low overlap regimes and large time horizons. The results from Theorem 1 are confirmed in Table 2+3 using synthetic data.
> >
> >     * **Action:** We have expanded the discussion of our experimental results and added the above insights.
> >
> > We thank you again for your valuable feedback and are happy to answer any remaining questions or to incorporate additional suggestions to improve the clarity and impact of our work.
> >
> >
> > ## References
> >
> > [1] Curth, van der Schaar (2021). Nonparametric estimation of heterogeneous treatment effects: From theory to learning algorithms. AISTATS.
> >
> > [2] Alaa et al. (2023). Conformal Meta-learners for Predictive Inference of Individual Treatment Effects. NeurIPS.
> >
> > [3] Acharki et al. (2023). Comparison of meta-learners for estimating multi-valued treatment heterogeneous effects. ICML.
> >
> > [4] Kennedy (2023). Towards optimal doubly robust estimation of heterogeneous causal effects. Electronic Journal of Statistics.
> >
> > [5] Foster, Syrgkanis (2023). Orthogonal Statistical Learning. The Annals of Statistics.
> >
> > [6] Nie, Wager (2021). Quasi-oracle estimation of heterogeneous treatment effects. Biometrika.
> >
> > [7] Melnychuk et al. (2022). Causal transformer for estimating counterfactual outcomes. ICML.

---

> > > ### Comment · Reviewer_Mx4T · 2024-11-17
> > >
> > > I want to thank the authors for their response. I have increased my score based on the rebuttal.

---

> > > > ### Author Response · Authors · 2024-11-18
> > > > **Thank you**
> > > >
> > > > Thank you for your fast response to our rebuttal and for increasing your score. As promised, we will include all changes in the camera-ready version of our paper.

---

### Official Review · Reviewer_1hT1 · 2024-11-04

**Soundness:** 4
**Presentation:** 4
**Contribution:** 4
**Rating:** 8
**Confidence:** 2

**Summary:**

The authors study the problem of estimating heterogeneous treatment effects in a time-dependent setting, where the treatment and measured covariates can change at each discrete step of a pre-specified time interval. The authors consider several possible meta-learning strategies for conditional average potential outcome (CAPO) and conditional average treatment effect (CATE) estimation which can be combined with any machine learning estimate of a base nuisance function, and they provide a theoretical analysis of the asymptotic error rate of each method. Based on the limitations of the initially proposed meta-learners, they propose the inverse variance weighted doubly robust learner to reduce instabilities that may arise due to low data availability. Finally, they confirm their theoretical findings and insights on both synthetic and real data.

**Strengths:**

The problem under study is of high practical relevance. While there has been a great proliferation of data availability in recent years, most of it is observational in nature, and collecting data from strict randomized controlled trials is still expensive. Methods that can take advantage of such data are critical. The time-varying aspect of the problem is also a natural feature of observational data collected "in the wild," making the problem setting even more relevant.

The presentation of the technical results is excellent. I am not an expert on causal inference, and the bare statement of one of the main results of the paper (Theorems 1) is highly technical in nature. However, the authors provide helpful interpretations of their results which make the value of the theorem clear. This clarity of exposition can be seen in many other places in the paper:

- An explanation for the lack of equality in equation (3) (failure to adjust for post-treatment confounders).
- An explanation for the constant variance assumption in Theorem 2. I had planned to add a question about this assumption to my review, but the authors addressed it immediately.
- Insight on the weighting mechanism of the proposed inverse variance weighted doubly-robust learner (lines 413-419), especially showing the connection to the simpler but more familiar binary treatment and static (not time-varying) settings.
- Intuition for which methods should be expected to perform best on which synthetic datasets and why.

These additional remarks and insights make the results of the paper far more accessible than they would be otherwise.

The results themselves are also strong. The authors give a theoretical analysis of the asymptotic behavior of each of the studied methods and discuss the implications for when each method should be preferred in practice. They then confirm their theory with synthetic datasets whose qualitative differences mimic some of the possible variation one could encounter when applying these methods in practice. Finally, they also obtain improved performance on real world healthcare data, which is the primary motivating example for the problem under study.

**Weaknesses:**

While the intuitions that the authors have already provided are greatly appreciated, additional context for some of the quantities used in the paper would be helpful for a non-expert audience. For instance, additional explanation of the nuisance functions in an appendix would be helpful. Specific examples of the rate functions used in Theorem 1 for some particular settings would also make the theorem easier to parse.

**Questions:**

1. What is the precise definition for a regression estimator to admit a particular rate in Assumption 3?

2. The real-world experiment reports RMSE on predicted normalized blood pressure. What is the normalization, and should the results of the proposed method be considered "good" or "bad" in absolute terms (not just relative to the other methods, over which it clearly obtains improvement)? It would be helpful to include an analysis of this sort to indicate where there is still room for future work.

---

> ### Author Response · Authors · 2024-11-17
> **Response to Reviewer 1hT1**
>
> Thank you very much for your thoughtful review and positive evaluation of our paper! Below, we address your comments and suggestions in detail.
>
> ## Response to “Weaknesses”
>
> Based on your feedback, we have made the following improvements to the paper (see the changes in our paper in red color):
>
>
>
> * We **expanded Appendix E** to include more **details regarding nuisance parameters** and the methods used to implement the corresponding models for their estimation.
> * We added a **new Appendix H**, where we **formally define the rates** in Theorem 1 and provide explicit examples to make the results more accessible for a non-expert audience.
>
> These additions aim to improve the clarity and accessibility of the paper for a broader readership.
>
>
> ## Response to “Questions”
>
>
>
> 1. **Precise definition of rates**. Thank you for the suggestion. As outlined in our response to weaknesses, we added a **new Appendix H** where we formally define the rates in Theorem 1. This includes specific examples and additional context to make the assumption and its implications clearer.
> 2. **Normalization in real-world experiments.** Thank you for bringing up this important point. We used a standard normalization procedure, where we subtracted the mean $61.09$ and the standard deviation $14.48$. We would like to emphasize that our experiments aim to validate the theoretical results and compare the performance of different meta-learners relative to each other. As such, we do not claim absolute optimality of our results. There is indeed room for future work, such as combining our meta-learners with tailored state-of-the-art model architectures for specific datasets. Our paper lays the theoretical foundation for these future developments and provides practical guidelines for designing such algorithms.  \
> **Action:** We added details regarding the normalization to our paper. We also added a new discussion section, in which we outline possible future directions such as developing new models based on our meta-learners.

---

> > ### Comment · Reviewer_1hT1 · 2024-11-26
> >
> > Thanks to the authors for their response. I believe the additional clarifications are helpful. I am also encouraged that the other reviewers seem to have had their concerns addressed, so I will maintain my score.

---

### Official Review · Reviewer_ytb1 · 2024-11-08

**Soundness:** 3
**Presentation:** 3
**Contribution:** 3
**Rating:** 6
**Confidence:** 3

**Summary:**

This paper proposes model-agnostic meta-learners to estimate temporal HTE. Specifically, authors address the main challenges of temproal HTE estimation from a unified view. They theoretically analyze when specific learners are preferable over others. They derive inverse variance weights to stabilize the loss of the DR-learner.

**Strengths:**

1. Authors provided a comprehensive related works section.
2. The problem setting is clear and overall reasonable.
3. The theoretical analysis is good and comprehensive, although I can't fully check the correctness.
4. The empirical validation is comprehensive too.

**Weaknesses:**

1. I suggest authors explain their motivation in detail. I believe the current version "Despite the fact that meta-learners are often considered state-of-the-art in the static setting, similar learners for the time-series setting are missing."  is not enough. How exactly meta-learners benefit or fit the time-varying setting? I think the reason: "meta-learners are SOTA under static setting" is not convincing.

2.  The caption of Figure 1 can be more specific. I suggest authors add some explanation for the edges, for example, why X(t-2) has a direct causal influence on X(t+1)? What is the practical meaning of such causal path?

3. I think some important concepts are not clearly defined. For example, does response surfaces equal response functions? I presume they are the same estimands.

4. I understand the PI-RA-learner, which can be seen as the estimator version of Eq 5. However, can authors explain the pseudo-outcomes (Eq. 8) in detail? When At = at or bt, why the time index of history info (H) is t+1 rather than t?

5. The contribution of methodology may be thin, since I believe IPW estimator for heterogeneous treatment effects or counterfactual outcomes estimation is already proposed by Papadogeorgou et al (2022). 《Causal inference with spatio-temporal data: estimating the effects of airstrikes on insurgent violence in iraq.》 Journal of the Royal Statistical Society
Series B: Statistical Methodology, 84(5):1969–1999.

6. In eq.9, authors define the propensity scores in the time-varying setting. My concerns are about the implementation details about this propensity score. To be specific, the action (treatment) sequence Al, is a high-dimensional variable. So how do authors employ history info Hl to model the probability distribution of this high-dimensional variable? Please clarify my concerns.

7. In the experiment section, I understand authors try to emphasize that their proposed estimators do not rely on any specific model architecture. However, to implement an estimator, one have to choose a network, and I believe different network architectures can have diverse performances. In addition, there may be a best model (e.g. Transformer or Mamba). So I think the contribution of "MODEL-AGNOSTIC" may not be appropriate.

8. BTW, it seems that authors do not discuss the computation efficiency. I suggest authors add this section in the main text or Appendix.

**Questions:**

Please refer to "Weaknesses".

---

> ### Author Response · Authors · 2024-11-17
> **Response to Reviewer ytb1**
>
> Thank you for your review and your helpful comments. As you can see below, we have carefully revised our paper along with your suggestions.
>
>
> ## Response to “Weaknesses”
>
> 1. **Motivation for meta-learners.** We agree that the statement "meta-learners are state-of-the-art under the static setting" requires further elaboration. Causal inference is, at the fundamental level, a **statistical estimation problem with nuisance parameters**—HTEs cannot be estimated directly but depend on accurate estimation of response functions and propensity scores. The statistical theory underpinning such estimation problems builds on the **efficient influence function (EIF)** of the target estimand, resulting in model-agnostic estimators that are robust w.r.t. errors in the nuisance estimators [1, 2]. For finite-dimensional estimands (e.g., average treatment effects), these EIF-based estimators achieve efficiency bounds (i.e., are **provably optimal**) [1, 2]. For HTEs, recent work shows that they retain favorable properties as well (e.g., Neyman orthogonality and robust convergence rates) [3, 4, 5, 6, 7]. In the HTE literature, such model-agnostic learners are called **meta-learners**. Meta-learners are thus **well-motivated by established statistical theory and have become standard in causal inference** [e.g., 3, 4, 5, 6, 7]. For example, meta-learners are **implemented in commonly used software packages for HTE estimation** (e.g., EconML, DoubleML). In our paper, we close an important gap by extending and analyzing meta-learners for longitudinal data and time-series settings, and, thereby, we address challenges unique to these contexts.  \
> **Action:** We removed our statement "meta-learners are state-of-the-art under the static setting" and replaced it with a more detailed motivation.
> 2. **Clarifying Figure 1.** Thank you for pointing this out. We have added a concrete example to clarify why such edges might exist. For instance, consider heart rate or blood pressure measurements recorded over fine-grained time intervals. These measurements naturally exhibit temporal dependencies, with earlier states directly influencing later ones. We emphasize, however, that **our methods do not require such edges to exist**. In causal inference, **assumptions correspond to the absence of edges, not (!) their presence**.  \
> **Action:** We incorporated these clarifications into our problem setup.
> 1. **Response surfaces vs. response functions.** Thanks! You are correct, and we apologize for the ambiguity.  \
> **Action:** We have streamlined our terminology to consistently use "response functions" throughout the manuscript.
> 2. **RA-pseudo-outcome (Eq. 8).** You raise an important point. The time index of history information, $\bar{H}{t+1}$, ensures that the pseudo-outcome equals the CATE in expectation, i.e., ${\tau}^{\bar{a}, \bar{b}}(\bar{h}{t}) =\mathbb{E}[{Y}_\mathrm{RA}^{\bar{a}, \bar{b}} | \bar{H}{t} = \bar{h}{t}]$. We acknowledge that it is not immediately obvious why this is necessary.  \
> **Action:** To address this, we have added **additional proofs** in our **new Appendix C.2**, explicitly showing that all pseudo-outcomes considered in our paper are unbiased and can be used as second-stage regression targets.
> 3. **Contributions.** Thank you for pointing out the paper from Papadogeorgou et al. (2022). However, the paper focuses on a **different target estimand**—the *average* treatment effect of a stochastic policy (but not *heterogeneous* treatment effects). Furthermore, the proposed IPW learner is not our main contribution. Rather, we see our contributions in (i) the derivation of inverse-variance weights (Theorem 2) and (ii) the comprehensive theoretical analysis (Theorem 1), both of which are novel and specific to our setting.  \
> **Action:** We now cite Papadogeorgou et al. (2022) in our related work (Appendix B) and spell out clearly how our work is different and thus novel.
> 4. **Propensities with high-dimensional history.** You are correct that the propensity scores are defined with respect to the entire history of the respective time points, making them potentially high-dimensional. However, our meta-learners allow for **arbitrary machine learning models** to estimate the propensity scores, particularly those that handle high-dimensional data effectively. In our experiments, we used transformer-based architectures due to their known effectiveness in learning representations from high-dimensional time-series data. Specifically, we employed an encoder-only architecture with a causal mask to avoid lookahead bias. For further implementation details, we kindly point to Section 7.1 and Appendix E. **Action:** We expanded Appendix E and added additional details regarding the implementation of our nuisance functions. We are happy to answer any remaining questions you may have regarding our model choices.

---

> ### Author Response · Authors · 2024-11-17
> **Response to Reviewer ytb1 #2**
>
> 7. **(a) Model agnostic property.** Please allow us to clarify the term "model-agnostic." You are absolutely correct that, in practice, performance is affected by the choice of model, and there may be an optimal model for specific data. This choice generally depends on factors such as the underlying data-generating process or sample size (e.g., for small samples, linear models might be preferred). By "model-agnostic," we mean that our **meta-learners can be instantiated with arbitrary machine learning models, and our theoretical analysis does not depend on a specific model choice**. For instance, our inverse-variance weights (Theorem 2) stabilize the DR loss regardless of whether a transformer, LSTM, or another model is used. Furthermore, the learners yield the same asymptotic rates (Theorem 1). **Action**: We clarified this in our revised paper. \
>  \
> (b) **Regarding our experiments**: Our experiments aim to verify the theoretical results (e.g., Theorem 1) that characterize properties of our meta-learners, **not of the underlying models**. For this reason, we used the same transformer-based architecture for all our meta-learners because this ensures a fair comparison. Of note, this type of benchmarking is **standard in causal inference** [e.g., 4, 5, 6]. Furthermore, we would like to point to Appendix H, where **we provided additional experiments with an LSTM instead of a transformer** and observed consistent results.
> 8. **Computational efficiency.** Thanks for pointing this out. Let $R$ be the runtime of the underlying model (e.g., a linear model, transformer, etc…). The runtimes of the meta-learners are given in the following table and depend on how many models are required to be trained.  \
> **Action:** We added this to Appendix A. We also reported the exact runtime $R$ of our transformer-based implementation in Appendix E.
>
> | Meta-Learner      | Runtime Complexity           |
> |--------------------|------------------------------|
> | PIHA-learner       | $R$                         |
> | PIRA-learner       | $\tau \times R$             |
> | RA-learner         | $(\tau+1) \times R$         |
> | IPW-learner        | $2 \times R$                |
> | DR-learner         | $(\tau+2) \times R$         |
> | IVW-DR-learner     | $(\tau+2) \times R$         |
>
>
>
>
> ## References
>
> [1] Van der Vaart (2000). Asymptotic statistics. Cambridge University Press.
>
> [2] Van der laan, Rubin (2007). Targeted maximum likelihood learning. International journal of biostatistics.
>
> [3] Chernozhukov et al. (2018). Double/De-Biased Machine Learning for Treatment and Causal Parameters. Econometrics Journal.
>
> [4] Foster, Syrgkanis (2023). Orthogonal Statistical Learning. The Annals of Statistics.
>
> [5] Curth, van der Schaar (2021). Nonparametric estimation of heterogeneous treatment effects: From theory to learning algorithms. AISTATS.
>
> [6] Kennedy (2023). Towards optimal doubly robust estimation of heterogeneous causal effects. Electronic Journal of Statistics.
>
> [7] Nie, Wager (2021). Quasi-oracle estimation of heterogeneous treatment effects. Biometrika.

---

### Author Response · Authors · 2024-11-17
**Response to all reviewers**

# Response to all reviewers

Thank you very much for the constructive evaluation of our paper and your helpful comments! We addressed all of them in the comments below. We also uploaded a **revised version of our paper**, where we highlight key changes colored in $\color{red}\textbf{red}$.

Following the feedback of the reviewers, our **main improvements** are the following:



* We added **additional proofs (new Appendix C2)** showing that all pseudo outcomes considered in our paper are unbiased for CATE/ CAPO when used as a second-stage regression target. We also elaborated on the motivation behind certain pseudo-outcomes such as the RA-outcome in Eq.(8).
* We added **explanations regarding our experimental setup**. In particular, we now point more directly to our **Appendix H, in which we repeated experiments with a different model architecture** to show the robustness of our results w.r.t. model choice. Furthermore, we elaborated on our definition of “model-agnostic” and **how our experimental setup follows established causal inference literature**.
* We **restructured parts of our paper and improved the writing**. In particular, we moved parts of our model description to the appendix and added a new discussion at the end of the paper, including limitations and an outlook on future work. Furthermore, we fixed several typos and grammatical errors.

We will incorporate all changes into the camera-ready version of our paper. Given these improvements, we are confident that our paper will be a valuable contribution to the causal machine learning literature and a good fit for ICLR 2025.

---

### Meta-Review · Area_Chair_iivK · 2024-12-19

**Metareview:**

**Summary**: The paper proposes new meta-learners for time-varying settings. Time-varying confounding makes statistical estimation of heteregeneous treatment effects (HTE) very challenging due to exacerbated issues of overlap, justifying the need for estimators that can appropriately account for bias-variance tradeoffs in finite samples. Drawing on existing literature on meta-learners in the static setting, the paper develops new meta/dr-estimators and uses standard techniques to show consistency of proposed estimators. Empirical evaluation on synthetic and one real data demonstrate reasonable performance on heterogeneous treatment effect estimation.

**Strengths**:
1. HTE in time-varying settings is an important problem and prior work has so far not discussed robust estimators using sequential ML models as nuisance estimators in full detail. Therefore reviewers found the contribution useful to the community.
2. The theoretical results, while drawing on standard literature in the area was considered an important advance.
3. The paper is overall well written and the results are interpreted for a broad audience.
4. Paper contributes both theoretical and empirical insights for doubly-robust meta-learners.

**Weakness**:
1. Empirical evaluation is very weak considering the limited evaluation in real world data
2. Several claims were not precisely explained, specifically around runtime efficiency, some writing flaws, no conclusion section, etc. Reviewers pointed out many of these issues and were subsequently addressed in the revisions.
3. My read on the author response on discussion and interpretation of real world empirical performance of the proposed estimators was weak in the rebuttal response and mainly provided a generic response as opposed to concrete ablations on possible reasons of why a specific estimator dominates despite theoretical demonstration of bias.
4. Interesting that authors remove sample splitting in empirical evaluation, I understand the temptation for longitudinal data but the claim that this is not necessary for experiments but only necessary for the theory to go through is a bit weird, and it could change confidence intervals. Nonetheless authors may want to temper the claim on Donsker class and choice of sample-fitting vs not or provide a very concrete citation in justification of these choices.

**Justification**:
After seeing the updates to the main draft, and the responses to the reviewer concerns, the manuscript is definitively stronger, and crisper. Reviewers also improved their scores based on author response and most of their concerns were addressed modulo empirical evaluations. Considering the strengths and weakness of the paper, while I think statistically the contribution is somewhat incremental, it is important for the community and therefore I am recommending an accept.

**Additional Comments On Reviewer Discussion:**

No major points were raised during discussion despite calls to reach a better consensus. My final recommendation is based on rebuttal responses, draft updates, and changes to reviewer course at the end of the rebuttal period.

---

### Decision · Program_Chairs · 2025-01-22

Accept (Poster)